# The claustrum-prelimbic cortex circuit through dynorphin/κ-opioid receptor signaling underlies depression-like behaviors associated with social stress etiology

Yu-Jun Wang[1,2,3,9], Gui-Ying Zan[1,2,9], Cenglin Xu [4,9], Xue-Ping Li[1,9], Xuelian Shu[1,2,9], Song-Yu Yao[5], Xiao-Shan Xu[6], Xiaoyun Qiu[4], Yexiang Chen[7,8], Kai Jin[6], Qi-Xin Zhou[6], Jia-Yu Ye[4,8], Yi Wang [4], Lin Xu [6] ✉, Zhong Chen [4] ✉ & Jing-Gen Liu [1,2,4,8] ✉

Ample evidence has suggested the stress etiology of depression, but the underlying mechanism is not fully understood yet. Here, we report that chronic social defeat stress (CSDS) attenuates the excitatory output of the claustrum (CLA) to the prelimbic cortex (PL) through the dynorphin/κ-opioid receptor (KOR) signaling, being critical for depression-related behaviors in male mice. The CSDS preferentially impairs the excitatory output from the CLA onto the parvalbumin (PV) of the PL, leading to PL micronetwork dysfunction by disinhibiting pyramidal neurons (PNs). Optogenetic activation or inhibition of this circuit suppresses or promotes depressive-like behaviors, which is reversed by chemogenetic inhibition or activation of the PV neurons. Notably, manipulating the dynorphin/KOR signaling in the CLA-PL projecting terminals controls depressive-like behaviors that is suppressed or promoted by optogenetic activation or inhibition of CLA-PL circuit. Thus, this study reveals both mechanism of the stress etiology of depression and possibly therapeutic interventions by targeting CLA-PL circuit.

Major depressive disorder (depression) is a psychiatric disorder characterized by the abnormal mood/emotion, cognition and behavior, seriously affecting human physical and mental health[1,2]. The high socioeconomic burdens and fairly limited therapeutic effects of current monoamine-based agents raise an urgent need to better understand the aetiology of depression so that new and more effective, fast-acting drugs can be developed. Multiple lines of evidence point to a strong link between psychological stress and psychiatric disorders[3,4]. Stress is a major predisposing factor for depression[5,6]. Despite extensive research, the neurobiological

[1]CAS Key Laboratory of Receptor Research and State Key Laboratory of Drug Research, Shanghai Institute of Materia Medica, Chinese Academy of Sciences, No. 555 Zuchongzhi Road, Shanghai 201203, China. [2]University of Chinese Academy of Sciences, No. 19 A Yuquan Road, 100049 Beijing, China. [3]Shandong Laboratory of Yantai Drug Discovery, Bohai Rim Advanced Research Institute for Drug Discovery, Yantai 264117, China. [4]Key Laboratory of Neuropharmacology and Translational Medicine of Zhejiang Province, School of Pharmaceutical Sciences, Zhejiang Chinese Medical University, Hangzhou, China. [5]School of Chinese Materia Medica, Nanjing University of Chinese Medicine, Nanjing 210023, China. [6]Laboratory of Learning and Memory, Key Laboratory of Animal Models and Human Disease Mechanisms, Kunming Institute of Zoology, Kunming 650223, China. [7]School of Basic Medicine and Clinical Pharmacy, China Pharmaceutical University, Nanjing 210009, China. [8]Department of Neurobiology and Acupuncture Research, The Third Clinical Medical College, Zhejiang Chinese Medical University, Key Laboratory of Acupuncture and Neurobiology of Zhejiang Province, Hangzhou 310053, China. [9]These authors contributed equally: Yu-Jun Wang, Gui-Ying Zan, Cenglin Xu, Xue-Ping Li, Xuelian Shu. ✉e-mail: lxu@vip.163.com; chenzhong@zju.edu.cn; jgliu@simm.ac.cn

mechanisms underlying stress-induced depression remains poorly understood.

Compelling evidence proposes a hypothesis that depression is linked to disruption of homeostatic mechanisms of cortical micronetwork that are mainly composed of excitatory pyramidal neurons (PNs) and inhibitory interneurons. Maintaining the stability of cortical micronetwork (excitation/inhibition balance, E/I balance) is critical for a range of higher-order brain functions and control of behaviors[7-11]. The homeostatic control of cortical micronetwork function is dependent on neural circuits constituted by the cortico-subcortical or cortico-cortical areas, which may be disrupted by stress through altering structural and functional plasticity of synapses in these circuits[12].

Glutamatergic neurons are the major excitatory neurons in the brain of mammals[13-15]. A remarkable characteristic of glutamatergic synapses is their ability to alter structure and function in response to environmental stimuli[16,17]. Because a vast majority of excitatory neurons in brain areas and circuits that are involved in mediating cognitive-emotional behaviors use glutamate as neurotransmitter, the glutamatergic neurons would be primary targets for stress and serve as key mediator of pathophysiology of depression[18,19]. The medial prefrontal cortex (mPFC), a key cortical region involved in modulation of cognitive and emotional behaviors, is a primary target of chronic stress[15,20,21] and displays a pivotal role in neural circuits underlying depression[12,22,23]. Chronic stress leads to spine loss and dendritic atrophy[24-26] and high excitability of glutamatergic neurons within the mPFC[10,27,28]. However, the molecular and circuit mechanisms by which stress induces disruption of cortical micronetwork remains poorly understood.

The mPFC receives excitatory projections from multiple subcortical areas including the claustrum (CLA), a poorly understood subcortical region[29]. The CLA sends most densely glutamatergic afferents to the PFC[30,31]. Distinct from other long-range excitatory inputs to the PFC, CLA neurons send biased excitatory input to inhibitory interneurons compared to excitatory PNs in the PFC, thus providing stronger feedforward inhibition (FFI) to cortical micronetworks[32,33]. Additionally, the CLA displays most highly expressing κ-opioid receptors (KORs)[34,35]. KOR signaling is a crucial mediator of glutamatergic transmission[36-38] and an important substrate for depression[39,40]. These findings lead to the hypothesis that the CLA might be likely to play a unique role in homeostasis control of the mPFC micronetworks and potentially implicated in chronic stress-induced depression through the neural circuits constituted by the CLA and the mPFC. Nevertheless, chronic stress-induced abnormalities in glutamatergic signaling in mPFC micronetwork through KOR signaling-mediated dysfunction of the CLA-mPFC circuit has not been observed yet.

This study employing a combination of anatomical tracing, pharmacological, optogenetic and chemogenetic manipulations and electrophysiological recordings unveiled a circuit-based framework wherein chronic social defeat stress (CSDS) evoked impairment of excitatory synaptic transmission from the CLA to the PL by potentiation of presynaptic dynorphin/KOR signaling, which resulted in biased inactivation of parvalbumin-positive (PV) neurons in the PL, thereby leading to high excitability of PL PNs and depressive-like behaviors via disinhibiting pyramidal neurons.

## Results

### The CLA plays an important role in depressive-like behaviors
As we identified that ibotenic acid lesion of the CLA enhanced depressive-like behavior (Supplementary Fig. 1), in the first set of experiments we used CSDS paradigm to model depressive-like behavior in mice[41]. We employed CSDS model in combination of chemogenetics to manipulate CLA glutamatergic neuronal activity in a cell-type specific manner[42]. Mice were subjected to 10-days CSDS or sub-threshold social defeat stress (SSDS) (Supplementary Fig. 2). CSDS

mice displayed a robust depressive phenotype marked by a significant reduction in social interaction (SI) ratio and the time spent in the interaction zone (Supplementary Fig. 2a-c). Mice that underwent SSDS did not show depressive-like behavior (Supplementary Fig. 2d-f), but they were more vulnerable to subsequent stress established in the literature[43]. To selectively inhibit or activate CLA glutamatergic neurons, we microinjected the AAV expressing mCherry-tagged inhibitory Gi DREADD (hM4Di) or excitatory Gq DREADD (hM3Dq) under the control of the CaMKIIα promoter into the CLA and achieved inhibition or activation of CLA glutamatergic neurons by intraperitoneal (i.p.) injection of clozapine-N-oxide (CNO, 2.5 mg/kg), an inert ligand specific to hM4Di and hM3Dq receptors (Fig. 1a, b, h, i).

To examine whether mCherry⁺ neurons in the CLA are glutamatergic neurons, we performed in situ hybridization for examining co-localization of mCherry with the marker of glutamatergic neuron (vGlut1 mRNA). We found that most (85%) mCherry⁺ neurons co-expressed with vGlut1 mRNA, with only a small fraction (9%) expressing inhibitory GABAergic marker (GAD65) (Supplementary Fig. 3a, b). Next, we performed electrophysiological recordings in CLA slices to validate the inhibitory and excitatory effect of the hM4Di and hM3Dq on CLA glutamatergic neuronal activity. Bath application of CNO (10 μM) significantly inhibited and facilitated the firing of action potentials in hM4Di- and hM3Dq-expressing CLA glutamatergic neurons, respectively (Supplementary Fig. 3c, d). We then examined the effects of manipulating CLA glutamatergic neurons with chemogenetics on depressive-like behaviors in mice. We found that mice that were subjected to SSDS and received chemogenetic inhibition by Gi DREADD exhibited depressive phenotypes, as evidenced by decreased SI ratio and the time spent in the interaction zone in social interaction test (SIT), decreased sucrose preference in the sucrose preference test (SPT) and increased immobility in the tail suspension test (TST) (Fig. 1c-f). Moreover, chemogenetic inhibition of CLA glutamatergic neurons also induced depressive-like behaviors in control mice (Supplementary Fig. 3e-i). These data suggest that inhibition of CLA glutamatergic neurons promotes susceptibility to depression. In contrast to the pro-depressant effect of chemogenetic inhibition of CLA glutamatergic neurons, chemogenetic activation of CLA glutamatergic neurons reversed CSDS-induced depressive phenotypes, as indicated by increased time spent in the interaction zone and SI ratio in the SIT, increased sucrose preference in the SPT and decreased immobility in the TST (Fig. 1j-m). Neither chemogenetic activation nor inhibition of CLA glutamatergic neurons had any effect on locomotor activity (Fig. 1g, n). Taken together, these results suggest that CLA glutamatergic neurons are critically involved in depressive-like behaviors associated with CSDS.

### CSDS impairs excitatory output of the CLA to the PL, but evokes conversely changes in activity of PNs and PV neurons within the PL
Next, we sought to determine neuronal circuits that are involved in CLA regulation of depressive-like behaviors. The CLA is largely composed of glutamatergic neurons, with only 10%-15% of GABAergic interneurons[44]. To determine the targets of claustral excitatory afferents, we injected AAV-CAMKIIα-mCherry, a viral-mediated anterograde tracer, into the CLA of mice, which resulted in robust labeling of several downstream regions, such as the PL, ventral pallidum (VP), basolateral amygdala (BLA) and the mediodorsal thalamic nucleus (MD) (Supplementary Fig. 4a). The projections from the CLA to PL were further confirmed by the injection of the retrograde tracer Fluoro-Gold into the PL (Supplementary Fig. 4b). Based on that our recent study showed PL participating in CSDS-induced depressive-like behaviors[45] and that viral- and Fluoro-Gold-mediated tracing demonstrated that the CLA glutamatergic neurons projected to the PL, we asked whether the CLA-PL circuit was critically involved in CSDS-induced depressive-like behaviors.

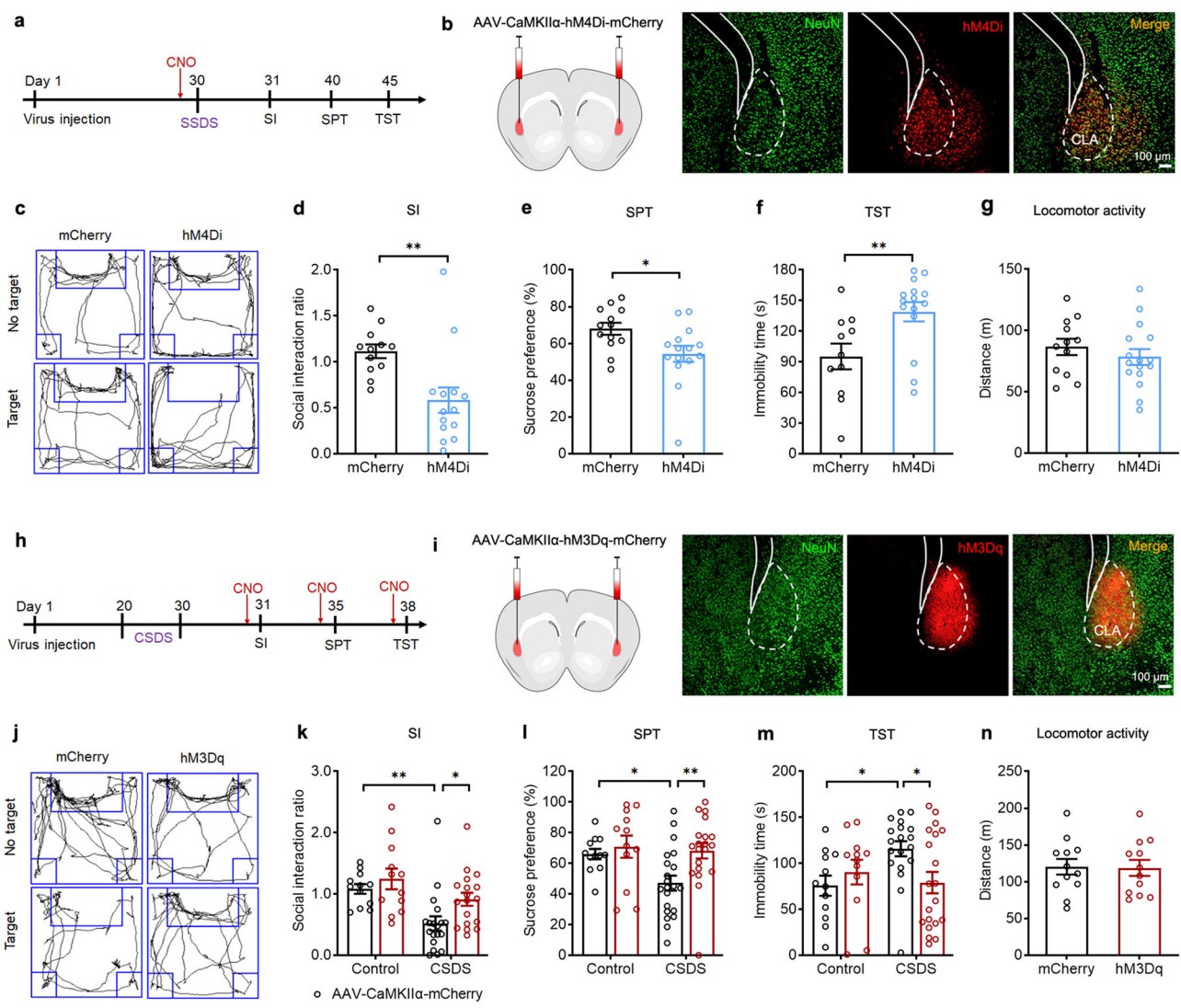

**Fig. 1 | CLA glutamatergic neurons are crucially involved in the regulation of depressive-like behaviors associated with stress. a**, **h** Schematic diagram of experimental procedure. CNO was pretreated for 40 min before SSDS (**a**) or before each test (CSDS). Left: schematic of injection AAV-CaMKIIα-hM4Di-mCherry (**b**) or AAV-CaMKIIα-hM3Dq-mCherry (**i**) into CLA. Right: representative images showing express hM4Di-mCherry (**b**) and hM3Dq-mCherry (**i**) in CLA. Scale bar, 100 μm. **c**, **j** Representative exploration tracks in the social interaction test. **d**–**g** Chemogenetic inhibition of CLA glutamatergic neurons in mice underwent SSDS decreased social interaction ratio (**d** $n = 12$–14, $t_{(24)} = 3.231$, $P = 0.0036$. Student's $t$ test), decreased sucrose consumption in the SPT (**e** $n = 13$-15, $t_{(26)} = 2.437$, $P = 0.0219$. Student's $t$ test) and increased immobility time in the TST (**f** $n = 11$–15,

$t_{(24)} = 2.830$, $P = 0.0093$. Student's $t$ test), without affecting locomotor activity (**g** $n = 12$–16, $t_{(26)} = 0.8715$, $P = 0.3914$. Student's $t$ test) compared to the mCherry control group. **k**–**n** Chemogenetic activation of CLA glutamatergic neurons in mice underwent CSDS increased social interaction ratio (**k** $n = 12$–19, $F_{(1,57)} = 0.8662$, $P = 0.3559$. Two-way ANOVA), increased sucrose consumption in the SPT (**l**, $n = 12$-21, $F_{(1, 61)} = 2.196$, $P = 0.1435$. Two-way ANOVA) and decreased immobility time in the TST (**m** $n = 12$–20, $F_{(1, 59)} = 5.117$, $P = 0.0274$. Two-way ANOVA), without affecting locomotor activity (**n** $n = 12$, $t_{(22)} = 0.09723$, $P = 0.9234$. Student's $t$ test) compared to the mCherry control group. All data are shown as mean ± s.e.m. *$P < 0.05$, **$P < 0.01$. Source data are provided as a Source Data file.

To address this, we initially determined whether CSDS triggered the CLA-PL circuit-level maladaptation in neuronal activity and synaptic transmission. PL micronetworks are composed of excitatory PNs and inhibitory GABAergic interneurons. Given that CLA glutamatergic neurons project to both PNs and GABAergic interneurons in the PFC[32] and that PV interneurons represent the largest group of GABAergic interneurons[46] and play an important role in stress-induced disorders[10,28,47], we detected the effects of CSDS on the activity of glutamatergic neurons in the CLA and the PL as well as PV neuron in the PL by measurement of c-fos expression. We found that CSDS resulted in a significant decrease of the co-localization of c-fos with excitatory neurons in the CLA (Fig. 2a, b) and the co-localization of c-fos with PV neurons in the PL (Fig. 2m, n). However, CSDS led to a significant

increase of the co-localization of c-fos with excitatory neurons in the PL (Fig. 2g, h). The results suggest that CSDS decreases glutamatergic neuronal activity in the CLA and PV neuronal activity in the PL, but enhances glutamatergic neuronal activity in the PL. Next, we determined whether CSDS altered the functions of CLA glutamatergic neurons and PL PV interneurons by electrophysiological recordings. To this end, we injected AAV-CaMKIIα-ChR2-EYFP into the CLA to label PL-projecting CLA glutamatergic neurons and AAV-EF1α-DIO-mCherry into the PL of PV-Cre mice to label PV interneurons. Using patch-clamp recordings from brain slices containing the CLA prepared from CSDS mice, we found that CSDS reduced the excitability of CLA glutamatergic neurons, as indicated by elevating the minimum current required to elicit an action potential (Supplementary Fig. 5a, b)

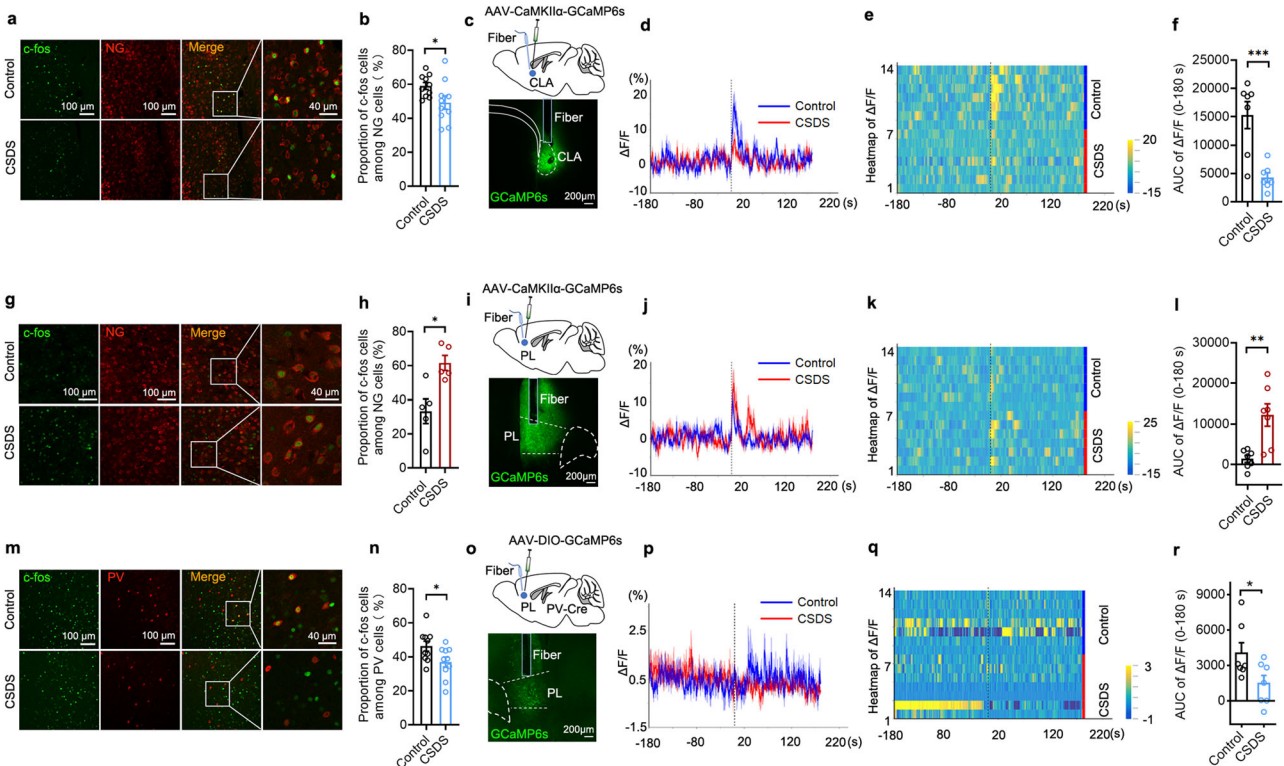

**Fig. 2 | CSDS decreased both CLA glutamatergic neuronal activity and PL PV interneuron activity, but increased PL glutamatergic neuronal activity.** CSDS decreased c-fos expression in CLA glutamatergic neurons (**a**, **b**) and decreased Ca²⁺ signals of CLA glutamatergic neurons (**c–f**). Representative image (**a**) and quantification of c-fos cells among CLA pyramidal neurons with neurogranin (NG) as markers in control and CSDS mice (**b** $n = 10$, $t_{(18)} = 2.203$, $P = 0.0409$. Student's $t$ test). **c** Schematic of virus injection and fiber implantation in the CLA. **d**, Representative traces of Ca²⁺ signals of CLA glutamatergic neurons in control mice (blue line) and CSDS mice (red line) during the social interaction test (−180s —— +180 s). The dashed line shows the introduction of CD-1mice. The curves and shaded regions indicate the mean ± s.e.m. $n = 7$ mice. **e** Heatmaps across animals aligned to the social interaction test. **f** Quantification of area under the curve (AUC, 0–180 s) of average ΔF/F during the social interaction test with CD-1 mice. ($n = 7$, $t_{(12)} = 4.355$, $P = 0.0009$. Student's $t$ test). CSDS increased c-fos expression in PL glutamatergic neurons (**g**, **h**) and increased Ca²⁺ signals of PL glutamatergic neurons (**i–l**). Representative images (**g**) and quantification of c-fos cells among PL glutamatergic neurons in control and CSDS mice (**h** $n = 5$, $t_{(8)} = 3.349$, $P = 0.0101$. Student's $t$ test). **i** Schematic of virus injection and fiber implantation in the PL. **j** Representative traces of Ca²⁺ signals of PL glutamatergic neurons in control mice

(blue line) and CSDS mice (red line) during the social interaction test (−180s —— +180 s). The dashed line shows the introduction of CD-1 mice. The curves and shaded regions indicate the mean ± s.e.m. $n = 7$ mice. **k** Heatmaps across animals aligned to the social interaction test. **l** Quantification of area under the curve (AUC, 0–180 s) of average ΔF/F during the social interaction test with CD-1 mice. ($n = 7$, $t_{(12)} = 3.751$, $P = 0.0028$. Student's $t$ test). CSDS decreased c-fos expression in PL PV interneurons (**m**, **n**) and decreased Ca²⁺ signals of PL PV interneurons (**o–r**). Representative images (**m**) and quantification of c-fos cells among PL PV neurons in control and CSDS mice (**n** $n = 10$, $t_{(18)} = 2.251$, $P = 0.0371$. Student's $t$ test). **o** Schematic of virus injection and fiber implantation in the PL of PV-Cre mice. **p** Representative traces of Ca²⁺ signals of PL PV neurons in control mice (blue line) and CSDS mice (red line) during the social interaction test (−180s —— +180 s). The dashed line shows the introduction of CD-1mice. The curves and shaded regions indicate the mean ± s.e.m. $n = 7$ mice. **q** Heatmaps across animals aligned to the social interaction test. **r** Quantification of area under the curve (AUC, 0–180 s) of average ΔF/F during the social interaction with CD-1 mice. ($n = 7$, $t_{(12)} = 2.288$, $P = 0.0411$. Student's $t$ test). All data are shown as mean ± s.e.m. *$P < 0.05$, **$P < 0.01$, ***$P < 0.001$. Source data are provided as a Source Data file.

and decreasing the number of action potentials elicited by stepped injected currents (Supplementary Fig. 5c). Reduced number of CLA glutamatergic neurons that expressed c-fos, together with lower excitability, suggests that CSDS decreases excitatory transmission of CLA projections. In brain slices containing PL prepared from CSDS mice, we detected the spontaneous excitatory postsynaptic currents (sEPSCs) of PV neurons and found that the frequency of sEPSCs was decreased, but the amplitude of sEPSCs was not altered significantly (Supplementary Fig. 5d–f), consistent with reduction of excitatory transmission from CLA to PL PV neurons. Decrease in the frequency of sEPSCs, together with reduced number of PL PV interneurons that expressed c-fos, suggests that CSDS impairs the function of PL PV interneurons.

To further confirm attenuation of the activity of glutamatergic neurons in the CLA but potentiation of it in the PL in response to CSDS, we then used fiber photometry recording of the activity of GCaMP6s-expressing CLA glutamatergic neurons and PL glutamatergic neurons and PV interneurons in mice subjected to CSDS. We found that the

intensity of the calcium signals was significantly lower on CSDS mice for glutamatergic neurons in the CLA (Fig. 2c–f, Supplementary Fig. 6a–c) and for PV interneurons in the PL (Fig. 2o–r, Supplementary Fig. 6g–i), however, the calcium signal of PL glutamatergic neurons was significantly higher on CSDS mice (Fig. 2i–l, Supplementary Fig. 6d–f), when compared with control mice. The results support that CSDS decreases the activities of glutamatergic neurons in the CLA and PV neurons in the PL, and increases the activity of glutamatergic neurons in the PL. Together, these data suggest that CSDS may impair excitatory transmission from the CLA to PL PV neurons, leading to increase in the activity of PL PNs by disinhibition, induced by decreased PV neuronal activity in the PL.

## Optogenetic manipulations of the CLA-PL circuit bidirectionally regulate depressive-like behaviors in mice that undergo CSDS or SSDS

To explore whether the CLA-PL circuit was implicated in CSDS-induced depressive-like behaviors, we first examined the effect of

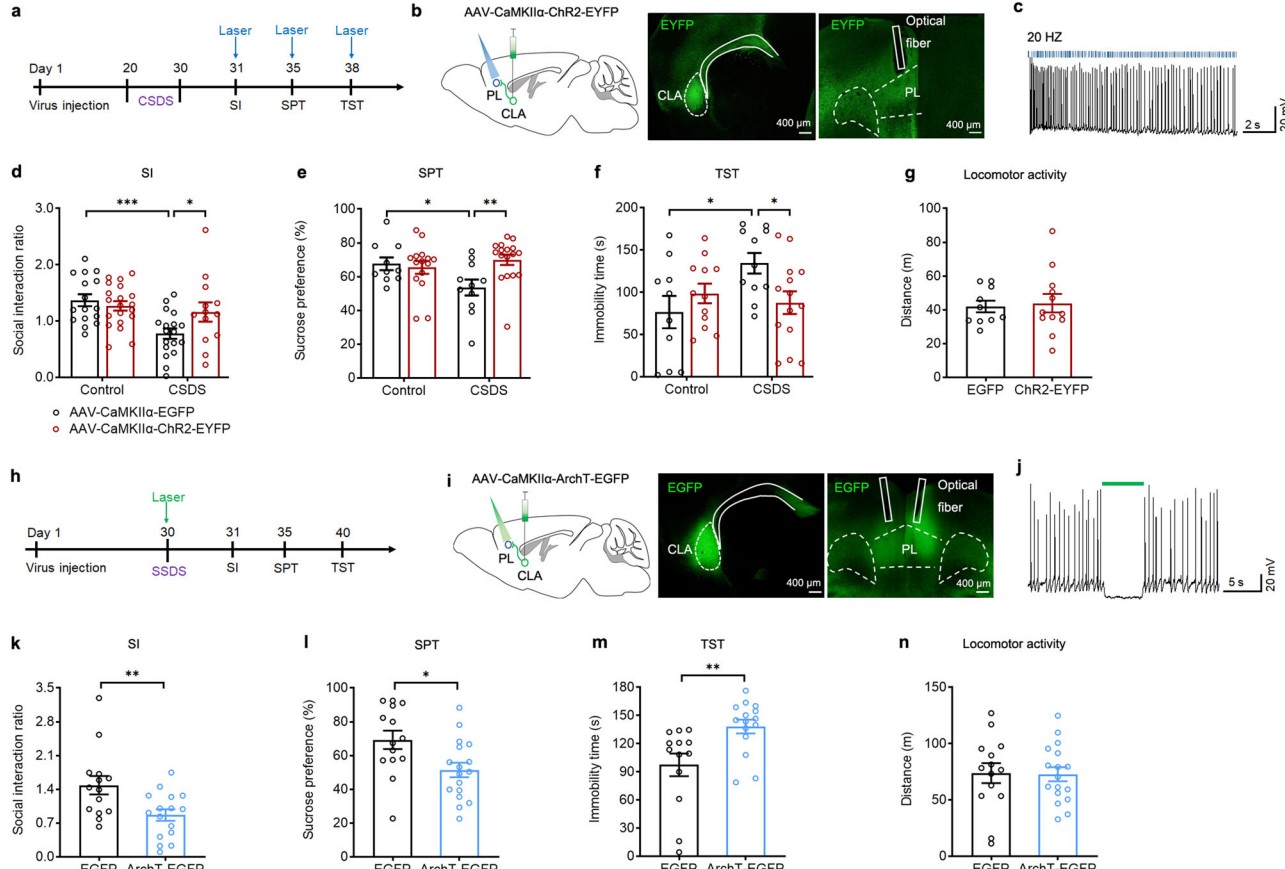

**Fig. 3 | Optogenetic activation or inhibition of PL-projecting CLA glutamatergic neurons bidirectionally regulates depression-like behaviors of mice.** Experimental timeline for optogenetic activation (**a**) or inhibition (**h**) of the CLA glutamatergic neurons. Left: schematic of CLA injection of AAV-CaMKIIα-ChR2-EYFP (**b**) or AAV-CaMKIIα-ArchT-EGFP (**i**) and PL implantation of optical fibers. Right: representative images showing EYFP or EGFP expression in both CLA and PL. Representative current clamp traces from in vitro recording of CLA ChR2-expressing neurons illuminated by 473 nm blue light stimulation (**c**) or ArchT-expressing neurons by 532 nm green light stimulation (**j**). Photoactivation of CLA^Glu-PL pathway reversed CSDS-induced decrease of social interaction ratio in the SIT (**d** $n = 13$–$20$, $F_{(1,63)} = 4.773$, $P = 0.0326$. Two-way ANOVA), decrease of sucrose

preference in the SPT (**e** $n = 10$–$17$, $F_{(1,49)} = 5.664$, $P = 0.0213$. Two-way ANOVA) and increase of immobility time in the TST (**f** $n = 10$–$14$, $F_{(1,43)} = 5.903$, $P = 0.0194$. Two-way ANOVA), without affecting locomotor activity (**g** $n = 10$–$12$, $t_{(20)} = 0.2825$, $P = 0.7805$. Student's $t$ test). **k**–**n** Photoinhibition of CLA^Glu-PL pathway in mice underwent SSDS decreased social interaction ratio in the SIT (**k** $n = 14$–$16$, $t_{(28)} = 2.808$, $P = 0.0090$. Student's $t$ test), decreased sucrose consumption in the SPT (**l** $n = 14$–$17$, $t_{(29)} = 2.605$, $P = 0.0143$. Student's $t$ test) and increased immobility time in the TST (**m** $n = 13$-$15$, $t_{(26)} = 2.976$, $P = 0.0062$. Student's $t$ test), without affecting locomotor activity (**n** $n = 14$–$17$, $t_{(29)} = 0.09209$, $P = 0.9273$. Student's $t$ test). All data are shown as mean ± s.e.m. *$P < 0.05$, **$P < 0.01$, ***$P < 0.001$. Source data are provided as a Source Data file.

disconnection of the CLA-PL circuit on depressive-like behaviors by unilateral lesion of the CLA and contralateral lesion of the PL using ibotenic acid, and found that the CLA-PL circuit disconnection resulted in depressive-like behaviors in mice subjected to SSDS (Supplementary Fig. 7). To confirm that the loss of excitatory transmission from the CLA to the PL is implicated in CSDS-induced depressive-like behaviors, we used optogenetics to specifically manipulate PL-projecting CLA glutamatergic neurons to verify the causal role of the CLA-PL circuit in depressive-like behaviors. If impaired activity of the CLA-PL circuit is a hallmark of CSDS leading to depression, strengthening the activity of this circuit could thus reverse depressive-like phenotypes associated with CSDS. To verify this, we expressed ChR2-EYFP or EGFP in the CLA glutamatergic neurons by bilaterally injecting the AAV-CaMKIIα-ChR2-EYFP or AAV-CaMKIIα-EGFP into the CLA and optical fiber was implanted over the PL to selectively stimulate the axonal terminals by turning on blue light (473 nm; 20 Hz, 5 mW, 5 ms) (Fig. 3a, b). Functional validation of ChR2-EYFP expression in CLA glutamatergic neurons was confirmed by using in vitro electrophysiological recordings (Fig. 3c). Optogenetic exciting of the CLA output terminals onto the PL almost completely rescued depressive-like behaviors to control level, without

affecting locomotion (Fig. 3d–g), suggests that the impaired CLA-PL excitatory drive by CSDS is a necessary condition for expressing depressive-like behaviors.

To further validate an essential role of impairment of excitatory transmission from the CLA to the PL in CSDS-induced depressive-like behaviors, we used the SSDS paradigm combined with optogenetic inhibition of the CLA-PL circuit. To this end, we bilaterally injected AAV-CaMKIIα-ArchT-EGFP or AAV-CaMKIIα-EGFP into the CLA, and optical fibers were implanted over the PL to selectively inhibit the CLA-PL circuit by illuminating a 532 nm green light into the PL (Fig. 3h, i). Functional validation confirmed that 532 nm laser reliably inhibited CLA neuronal firing in vitro (Fig. 3j). Behaviorally, we set the stimulation pattern as cycles of photoillumination of 9 s on with 1 s off, and found that optogenetic inhibition of the CLA-PL circuit of mice that underwent SSDS induced depressive phenotypes, as indicated by the reduction in SI ratio in the SIT and sucrose preference in the SPT and increase in immobility in the TST without change in locomotion (Fig. 3k–n). The results suggest that optogenetic silencing of the CLA excitatory output is also a sufficient condition for expressing depressive-like behaviors. Together, these data confirm the functional importance of the CLA-PL circuit in depression.

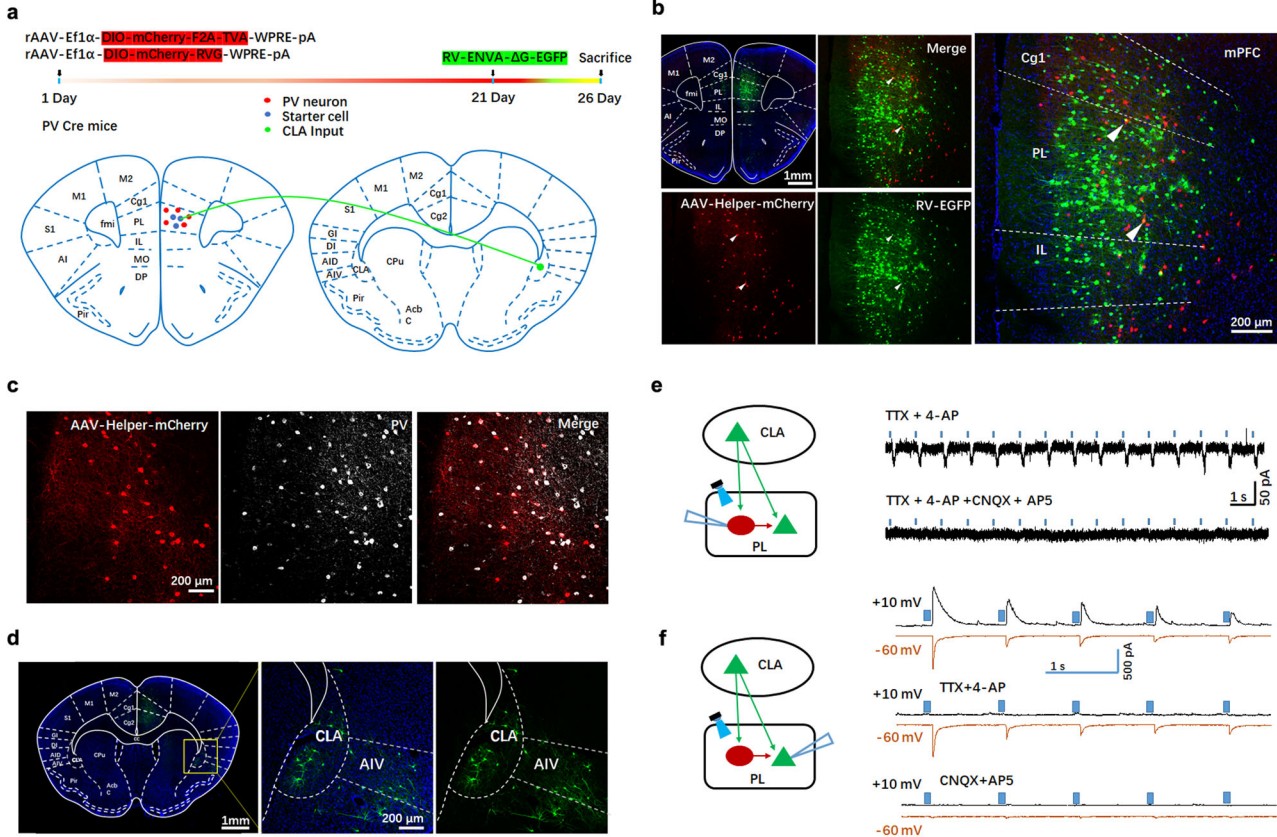

**Fig. 4 | PL PV interneurons receive direct projections from CLA glutamatergic neurons and yield FFI onto PL PNs. a** Timeline and schematic description for Cre-dependent retrograde trans-monosynaptic RV tracing experiments.
**b** Representative images of viral expression within PL of PV-Cre mice. Starter cells (yellow with arrowhead) co-express helper virus (AAV-Ef1α-DIO-mCherry-F2A-TVA-WPRE-pA, AAV-Ef1α-DIO-mCherry-RVG-WPRE-pA, red) and RV-ENVA-△G-EGFP (green). **c** Representative images showing colocalization of helper virus (red) with PV neurons (white). **d** Representative images showing expression of RV (green) within CLA. **e** Monosynaptic input from CLA glutamatergic neurons to PL PV interneurons. Left: a schematic showing the clamp recordings from PV inter-neurons in the PL in vitro while blue light was used to activate CLA ChR2-expressing axon terminals in the PL. Right: Representative monosynaptic responses of EPSCs in the presence of TTX, 4-AP and glutamatergic antagonists CNQX and AP5. **f** Activation of CLA glutamatergic neurons evoked EPSCs and IPSCs in PL gluta-matergic neurons. Left: a schematic showing the clamp recordings from gluta-matergic neurons in the PL in vitro while blue light was used to activate CLA ChR2-expressing axon terminals in the PL. Right: representative traces showing sIPSC (upper) and sEPSC (lower) recorded at +10 mV (upper) and −60 mV (lower) in the PL glutamatergic neurons. IPSCs of glutamatergic neurons was abolished after focal application of TTX and 4-AP while the EPSCs was not affected. EPSCs of gluta-matergic neurons was abolished after focal application of CNQX and AP5. Source data are provided as a Source Data file.

## PL-projecting CLA glutamatergic neurons exert a feedforward inhibitory effect on PL PNs through PL PV interneurons

In the PFC, PV interneurons and PNs are highly interconnected. The glutamatergic neurons from CLA synapse onto both PNs and PV interneurons within the PFC and can evoke potent FFI[32]. The afore-mentioned results demonstrated that CSDS resulted in a significant decrease of the activity of CLA glutamatergic neurons and PL PV interneurons, but a significant increase of PL PN activity, implying that CSDS may attenuate feedforward inhibitory effect of PV neurons on PNs in the PL by impairing excitatory transmission from the CLA to PL PV neurons. To investigate whether CSDS yielded depressive-like behaviors through the attenuation of feedforward inhibitory effect of PV neurons on PNs in the PL, we initially identified whether CLA glu-tamatergic neurons directly projected to PL PV interneurons by employing a cell-type-specific retrograde trans-monosynaptic tracing system[48,49]. Cre-dependent helper viruses (AAV-Ef1α-DIO-mCherry-F2A-TVA-WPRE-pA and AAV-Ef1α-DIO-mCherry-RVG-WPRE-pA) were injected into the PL of PV-Cre mice. After 3 weeks, the rabies virus (RV)-ENVA-△G-EGFP, which are unable to trans-synaptic spread, was injected into same site (Fig. 4a). The presence of these helper viruses can ensure trans-monosynaptic retrograde RV spread. We observed mCherry and EGFP double labeled neurons in the PL (Fig. 4b) and that mCherry signals were co-localized with the PV-specific antibody

signals (Fig. 4c), indicating that PL PV interneurons were labeled. EGFP-labeled neurons were also observed in the CLA (Fig. 4d), indicating that CLA neurons projected to PL PV interneurons.

We then confirmed whether CLA afferents synapsed directly onto PL PV interneurons and yield FFI of PNs in the PL using electro-physiological approaches. We injected AAV-CaMKIIα-ChR2-EYFP into the CLA of mice. Following optogenetic stimulation of CLA axons in the PL, we examined the CLA-evoked EPSCs (oEPSCs) in PV inter-neurons and found that optogenetic stimulation of CLA axons in the PL evoked oEPSCs in PV neurons. To examine whether CLA axons formed monosynaptic connections onto PL PV interneurons, we blocked action potential-dependent postsynaptic responses with bath appli-cation of TTX and potassium channel blocker 4-AP[50]. In PV cells tested, oEPSCs was still elicited by optical stimulation in the presence of 4-AP and TTX, indicating that CLA axons directly project onto PL PV neu-rons (Fig. 4e). Moreover, oEPSCs detected in PV interneurons were abolished by ionotropic glutamatergic antagonists CNQX and AP5, demonstrating that CLA afferents to PV interneurons are glutamater-gic transmission. Next, we determined whether PL PV neurons exerted FFI onto glutamatergic neurons in the PL. We recorded from gluta-matergic neurons in the PL in voltage clamp to isolate the excitatory and inhibitory currents elicited by stimulation of CLA afferents. Here, a cesium-based internal solution was used[51], and the excitatory inward

currents were measured with glutamatergic neurons held at −60 mV while inhibitory outward currents were recorded with glutamatergic neurons held at +10 mV (Fig. 4f). In addition to excitatory currents, inhibitory currents were also recorded in glutamatergic neurons of PL following optogenetic stimulation of CLA afferents (Fig. 4f), and these inhibitory currents were eliminated following bath application of TTX and 4-AP, demonstrating optical activating CLA inputs evoke polysynaptic inhibitory currents onto the glutamatergic neurons of PL through activating local inhibitory interneurons. Excitatory currents following optogenetic activation of CLA terminals were also blocked in the presence of CNQX and AP5, indicating that these responses represented were monosynaptic from CLA glutamatergic neurons. Together, these findings indicate that glutamatergic neurons in the PL receive FFI from PV interneurons as well as direct monosynaptic excitation driven by CLA afferents.

### Chemogenetic manipulations of PL PV neurons bidirectionally regulate optogenetic activation or inhibition of the CLA-PL circuit-mediated anti-or pro-depressant effects

Next, we determined whether the loss of inhibitory function of PV neurons induced by impaired excitatory inputs from CLA to PL PV neurons played a role in CSDS-induced depressive-like behaviors. We first employed pharmacological approach to test whether intra-PL administration of GABA receptor antagonists or agonists could reverse the anti- or pro-depressant effects of optogenetic activation or optogenetic inhibition of the CLA-PL circuit in mice that underwent CSDS or SSDS. Because PV interneurons belong to GABAergic interneurons, we thus examined the effect of intra-PL injection of a cocktail of GABA receptor antagonists (bicuculline+saclofen) on optogenetic activation of the CLA-PL circuit-induced antidepressant effect under the condition of CSDS. We found that GABA antagonists blocked optogenetic activation of the CLA-PL circuit-induced antidepressant effect without changing locomotor activity (Supplementary Fig. 8a–d). By contrast, intra-PL administration of a cocktail of GABA agonists (baclofen+muscimol) reversed optogenetic inhibition of the CLA-PL circuit -induced pro-depressant effect under the condition of SSDS without alteration of locomotor activity (Supplementary Fig. 8e–h).

To further confirm the essential role of dysfunction of PV neurons in CSDS-induced depressive-like behaviors, we combined optogenetic with chemogenetic approaches to determine whether chemogenetic inhibition or activation of PL PV neurons abolished anti- or pro-depressant effects caused by optogenetic activation or inhibition of the CLA-PL circuit. To this end, we injected AAV-CaMKIIa-ChR2-EYFP or AAV-CaMKIIa-ArchT-EGFP into the CLA and injected AAV-EF1a-DIO-hM4Di-mCherry or EF1a-DIO-hM3Dq-mCherry into the PL of PV-Cre mice, respectively (Fig. 5a, b, h, i). Optical fibers were implanted in the PL (Fig. 5b, i). The PV and mCherry double labeled neurons were observed in the PL of PV-Cre mice (Fig. 5c, j), indicating that hM4Di and hM3Dq were expressed in PV neurons. We also found that chemogenetic inhibition of PL PV neurons decreased optogenetic activation of the CLA-PL circuit-induced c-fos expression in the PL in CSDS mice (Supplementary Fig. 9a, b), whereas chemogenetic activation of PL PV neurons reversed optogenetic inhibition of the CLA-PL circuit-induced decrease in c-fos expression in the PL in SSDS mice (Supplementary Fig. 9c, d), confirming the effects of inhibitory hM4Di and excitatory hM3Dq on PL PV neurons and suggesting that PV neurons were affected by optogenetic activation or inhibition of the CLA-PL circuit. Accordingly, we also found that chemogenetic inhibition or activation of PL PV neurons reversed optogenetic activation or inhibition of the CLA-PL circuit-induced anti- and pro-depressant effects, as evinced by decreased or increased in SI ratio in the SIT, decreased or increased sucrose consumption in the SPT as well as increased or decreased immobility in the TST (Fig. 5d–f, k–m), but had no alteration on mice locomotor activity (Fig. 5g, n). Overall, these results suggest that

optogenetic activation or inhibition of the CLA-PL circuit-induced anti- or pro-depressant effects may be attributed to strengthening or suppressing PL PV interneuron activity. Taken together, these data support the necessity and sufficiency of PL PV interneuron activity for depressive-like behaviors.

### KORs negatively modulate CLA-PL synaptic transmission and critically participate in chronic stress-induced depressive-like behavior

KORs play a key role in regulation of negative emotions such as anxiety and depression[40,52] and are present in axon terminals of glutamatergic projections, providing inhibitory control over glutamatergic afferents[36,37]. Since our previous studies showed that the CLA displayed most highly expressing KORs in the brain of mouse and guinea pig[34,35], we sought to know whether KORs in the CLA were involved in CSDS-induced depressive-like behaviors by inhibiting excitatory synaptic transmission from the CLA to the PL. To this end, we first determined whether KORs were expressed in the CLA glutamatergic afferents to the PL and modulated excitatory synaptic transmission from the CLA to the PL. Results from double-labeling in situ hybridization of vGlut1 and KOR mRNA showed that 96% of KOR mRNA+ neurons in the CLA overlapped with vGlut1 mRNA and 45% of vGlut1 mRNA+ neurons overlapped with KOR mRNA (Supplementary Fig. 10a–c), suggesting that vGlut1 and KOR are co-localized in the CLA. We next determined whether KORs on CLA terminals participated in modulation of CLA-evoked optical EPSCs (CLA oEPSCs). We injected AAV-CaMKIIα-ChR2-EYFP into the CLA and AAV-EF1a-DIO-mCherry into the PL of PV-Cre mice and conducted whole-cell slice electrophysiology recordings from PV and glutamatergic neurons in the PL (Supplementary Fig. 10d, l). In vitro whole-cell electrophysiology recordings from PV interneurons and PNs in the PL showed that application of the KOR agonist U50, 488H reduced the amplitude of oEPSCs and simultaneously increased the paired pulse ratio (PPR) in both PV interneurons and PNs of brain slices containing PL prepared from naïve mice (Supplementary Fig. 10e–h, m-o) and both decreased amplitude of oEPSCs and increased PPR in PV neurons are larger than those of PNs. These data confirmed the presence of KORs on PL-projecting CLA glutamatergic neurons and suggesting that KORs negatively modulate synaptic transmission perhaps by inhibiting the probability of presynaptic glutamate release. The presence of KORs in the CLA glutamatergic afferents to the PL PV neurons was further supported by application of the KOR antagonist nor-BNI, which enhanced the excitatory inputs from the CLA to PV neurons (Supplementary Fig. 10i–k).

On the basis of the aforementioned results showing that KORs negatively modulated synaptic efficacy of CLA afferents to PL and impairment of excitatory transmission from the CLA to the PL contributed to depressive-like behaviors, we predicted that KORs in this circuit might be linked to CSDS-induced depressive-like behaviors. To address this, we first determined whether alteration of dynorphin/KOR signaling was occurred following CSDS. Western blot analysis showed that dynorphin, an endogenous KOR ligand, but not KOR expression in the PL of CSDS mice was significantly increased compared to that of control mice (Fig. 6a–d). It was also found that bilateral intra-PL administration of KOR antagonist nor-BNI before behavioral test blocked depressive-like behaviors (Supplementary Fig. 11a–c), supporting a key role of KORs in depression.

To confirm functional importance of KORs in modulation of depressive-like behaviors by regulating CLA afferents to the PL, we manipulated dynorphin/KORs signaling by activation and blockade of KORs with its agonist and antagonist and by alterations of KOR levels with overexpression and knockout during optogenetic activation and inhibition of the CLA-PL circuit. We first examined the effect of intra-PL injection of KOR agonist U50,488H on optogenetic activation of the CLA-PL circuit-induced antidepressant effect under condition of CSDS.

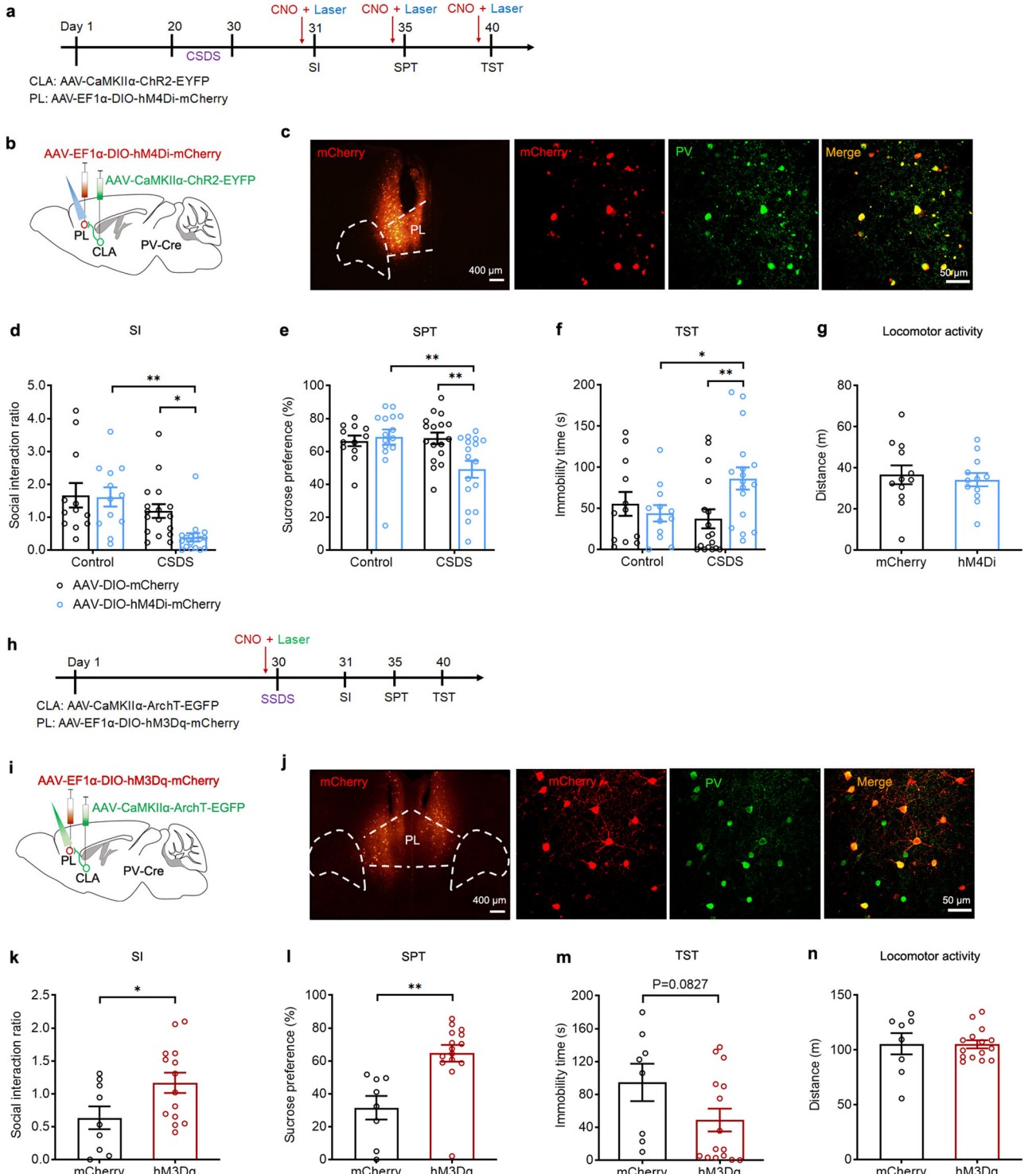

**Fig. 5 | Chemogenetic manipulations of PL PV neurons regulates the anti-or pro-depressant effects of optogenetic activation or inhibition of CLA$^{Glu}$-PL projection. a, h** Experimental timeline. **b, i** Schematic of CLA injection of AAV-CaMKIIα-ChR2-EYFP, PL injection of AAV-EF1α-DIO-hM4Di-mCherry and PL implantation of optical fiber (**b**) or CLA injection of AAV-CaMKIIα-ArchT-EGFP, PL injection of AAV-EF1α-DIO-hM3Dq-mCherry and PL implantation of optical fibers (**i**) in PV-Cre mice. **c, j** Representative images showing colocalization of mCherry positive cells with PV interneurons in the PL. **d–g** Chemogenetic inhibition of PL PV interneurons blocked photo-activation of the CLA$^{Glu}$-PL pathway induced increase of social interaction ratio (**d** $n = 12–17$, $F_{(1,54)} = 2.321$, $P = 0.1335$. Two-way ANOVA), increase of sucrose consumption in the SPT (**e** $n = 12–17$, $F_{(1,57)} = 5.784$, $P = 0.0194$.

Two-way ANOVA) and decrease of immobility time in the TST (**f** $n = 12–17$, $F_{(1,54)} = 5.477$, $P = 0.0230$. Two-way ANOVA) without affecting locomotor activity (**g** $n = 12$, $t_{(22)} = 0.4265$, $P = 0.6739$. Student's $t$ test). **k–n** Chemogenetic activation of PL PV interneurons reversed photo-inhibition of the CLA$^{Glu}$-PL pathway induced decrease of social interaction ratio (**k** $n = 9–14$, $t_{(21)} = 2.250$, $P = 0.0353$. Student's $t$ test), decrease of sucrose consumption in the SPT (**l** $n = 8–15$, $t_{(21)} = 3.800$, $P = 0.0010$. Student's $t$ test) and increase of immobility time in the TST (**m** $n = 8–15$, $t_{(21)} = 1.822$, $P = 0.0827$. Student's $t$ test) without affecting locomotor activity (**n** $n = 8–15$, $t_{(21)} = 0.05802$, $P = 0.9543$. Student's $t$ test). All data are shown as mean ± s.e.m. *$P < 0.05$, **$P < 0.01$. Source data are provided as a Source Data file.

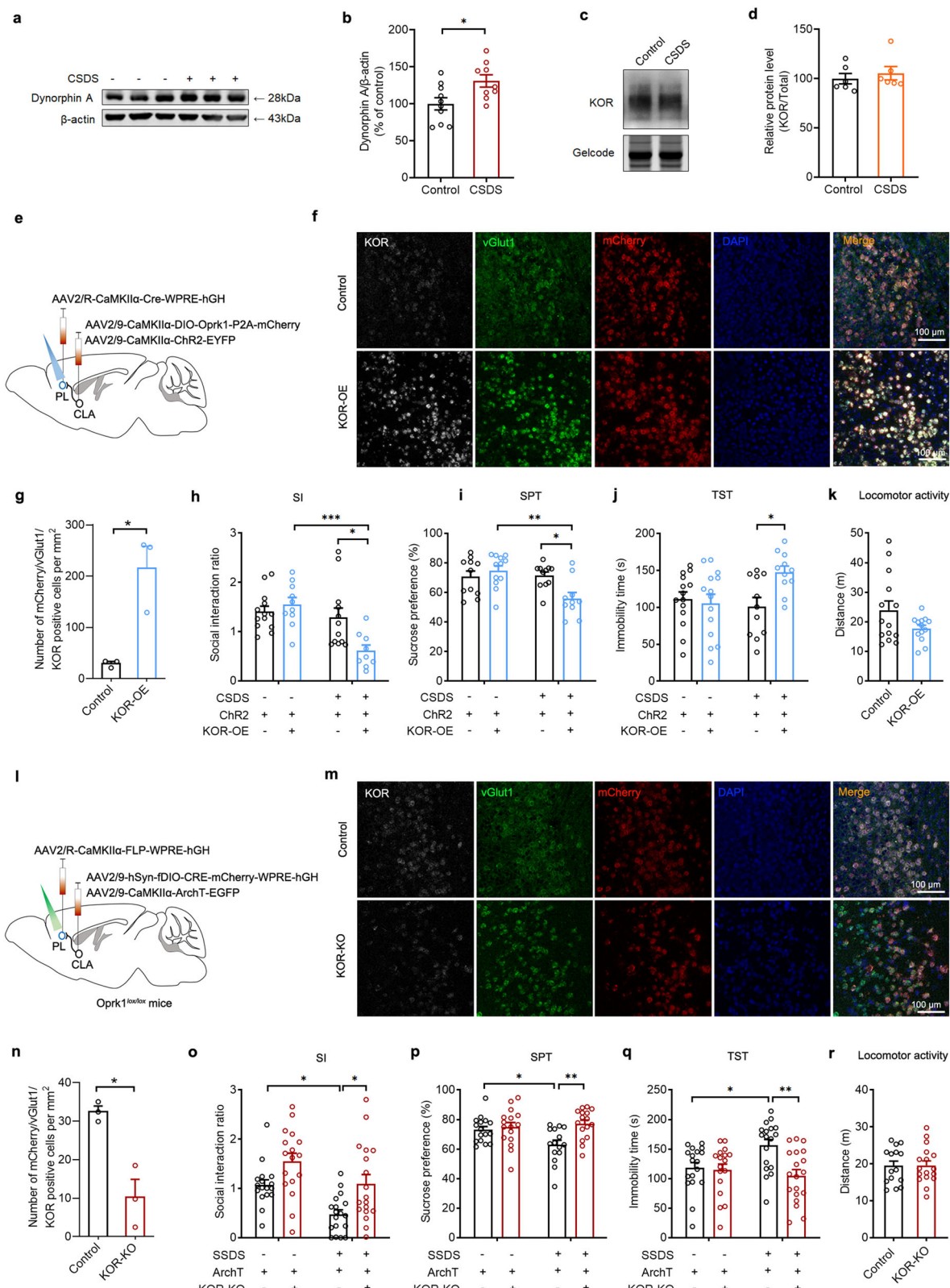

We found that U50,488H blocked optogenetic activation of the CLA-PL circuit-induced antidepressant effect (Supplementary Fig. 11d–f). By contrast, intra-PL administration of KOR antagonist nor-BNI reversed optogenetic inhibition of the CLA-PL circuit-induced pro-depressant effect under condition of SSDS (Supplementary Fig. 11h–j). Intra-PL administration of either U50,488H or nor-BNI has no effect on locomotor activity (Supplementary Fig. 11g, k).

To further confirm the role of KORs in PL-projecting CLA neurons in depression, we next manipulated KOR levels in PL-projecting CLA neurons by the strategies of circuitry specific overexpression and knockout. We first examined the effect of overexpression of KORs in PL-projecting CLA glutamatergic neurons on optogenetic activation of the CLA-PL circuit-induced antidepressant effect. We injected AAV2/9-CaMKIIα-DIO-Oprk1-P2A-mCherry and AAV2/9-CaMKIIα-ChR2-EYFP

**Fig. 6 | KORs in the CLA$^{Glu}$-PL circuit regulate depressive-like behaviors.**
**a** Representative immunoblots of dynorphin expression in the PL. **b** Quantification of dynorphin A ($n = 9–10$, $t_{(17)} = 2.655$, $P = 0.0167$. Student's $t$ test). **c** Representative immunoblots of KOR expression in the PL. **d** Quantification of KOR ($n = 6$, $t_{(10)} = 0.6318$, $P = 0.5417$. Student's $t$ test). **e, l** Schematic of CLA injection of AAV2/9-CaMKIIα-DIO-Oprk1-P2A-mCherry and AAV2/9-CaMKIIα-ChR2-EYFP and PL injection of AAV2/R-CaMKIIα-Cre-WPRE-hGH in WT mice (**e**) or CLA injection of AAV2/9-hSyn-fDIO-CRE-mCherry-WPRE-hGH and AAV2/9-CaMKIIα-ArchT-EGFP and PL injection of AAV2/R-CaMKIIα-FLP-WPRE-hGH in Oprk1$^{lox/lox}$ mice (**l**). **f, m** In situ hybridization of KOR (white), vGlut1 (green), mCherry (red) and DAPI (blue) in the CLA. **g, n** Quantification of KOR colocalization with vGlut1 and mCherry (**g** $n = 3$, $t_{(4)} = 4.288$, $P = 0.0128$; **n** $n = 3$, $t_{(4)} = 4.573$, $P = 0.010$. Student's $t$ test). **h–k** KOR over-expression in CLA$^{Glu}$-PL circuit blocked photo-activation of the CLA$^{Glu}$-PL

pathway induced increase of social interaction ratio (**h** $n = 9–13$, $F_{(1,40)} = 7.733$, $P = 0.0082$. Two-way ANOVA), increase of sucrose consumption in the SPT (**i** $n = 10–12$, $F_{(1,40)} = 8.265$, $P = 0.0064$. Two-way ANOVA) and decrease of immobility time in the TST (**j** $n = 11–14$, $F_{(1,46)} = 5.662$, $P = 0.0215$. Two-way ANOVA) without affecting locomotor activity (**k** $n = 13–14$, $t_{(25)} = 1.837$, $P = 0.0781$. Student's $t$ test). **o–r** knockdown of KOR in CLA$^{Glu}$-PL circuit reversed photo-inhibition of CLA$^{Glu}$-PL induced decrease of social interaction ratio (**o** $n = 17–18$, $F_{(1,66)} = 0.2481$, $P = 0.6201$. Two-way ANOVA), decrease of sucrose consumption in the SPT (**p** $n = 15–16$, $F_{(1,59)} = 5.132$, $P = 0.0272$. Two-way ANOVA) and increase of immobility time in the TST (**q** $n = 17–19$, $F_{(1,68)} = 5.837$, $P = 0.0184$. Two-way ANOVA) without affecting locomotor activity (**r** $n = 15–17$, $t_{(30)} = 0.06621$, $P = 0.9477$. Student's $t$ test). All data are shown as mean ± s.e.m. $^{*}P < 0.05$, $^{**}P < 0.01$, $^{***}P < 0.001$. Source data are provided as a Source Data file.

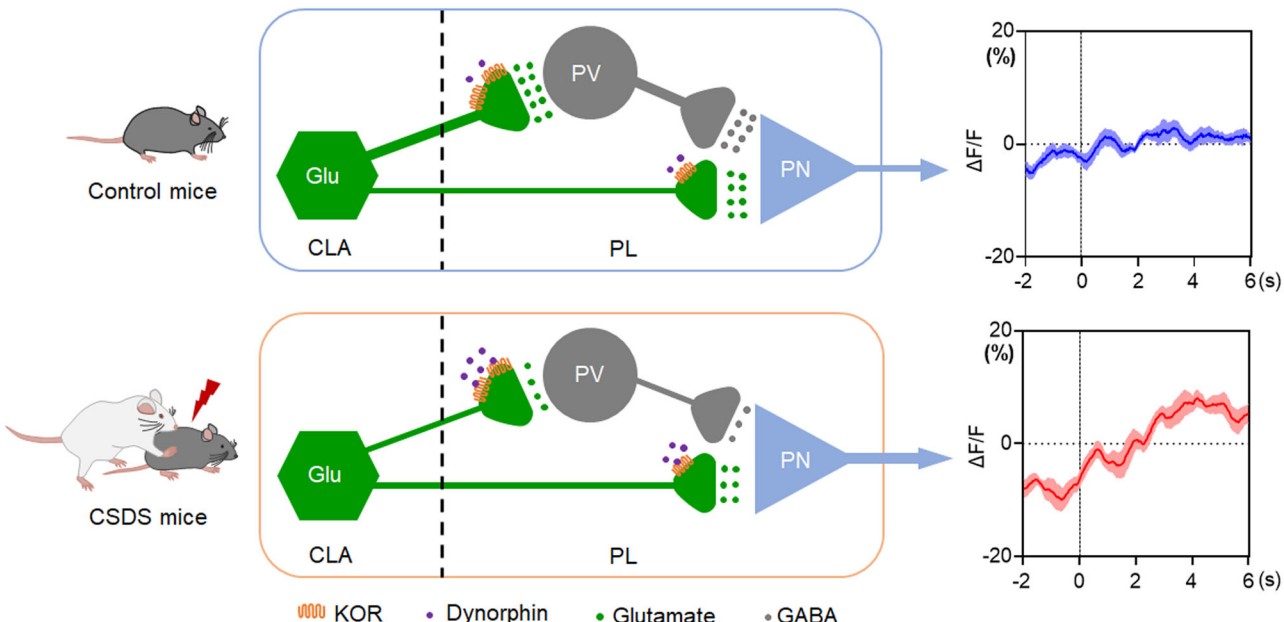

**Fig. 7 | Proposed model of circuit and molecular mechanism of depression associated with chronic stress.** Top panel, PL-projecting CLA glutamatergic neurons express KORs, synapse onto both excitatory pyramidal neurons (PNs) and PV interneurons in the PL and play a key role in the maintenance of the E/I balance in the PL through driving feedforward inhibition (FFI) of PNs by excitation of PV interneurons. Presynaptic KOR signaling profoundly inhibits the excitatory transmission from the CLA to PL PV neurons (Supplementary Fig. 10). Bottom panel, Chronic stress results in endogenous dynorphin release and increase KOR signaling in the PL. More strongly increasing KOR signaling of CLA synapses in PL PV interneurons lead to preferential impairment of excitatory transmission from the CLA to PL PV interneurons, which in turn induces the hyper-excitation of PL PNs as a result of the loss of FFI of PV interneurons on PNs in the PL, thereby resulting in E/I imbalance in the PL and depression. Source data are provided as a Source Data file.

into CLA and AAV2/R-CaMKIIα-Cre-WPRE-hGH into PL (Fig. 6e). Expressional validation of KORs was confirmed by in situ hybridization (Fig. 6f, g). As expected, overexpression of KORs in PL-projecting CLA glutamatergic neurons reversed optogenetic activation of the CLA-PL circuit-induced antidepressant effect, evinced by a significant decrease in SI ratio in the SIT and sucrose preference in the SPT and increase in immobility in the TST (Fig. 6h–j). Next, we examined the effect of knockdown of KORs in PL-projecting CLA glutamatergic neurons on optogenetic inhibition of the CLA-PL circuit-induced pro-depressant effect. We injected AAV2/9-hSyn-fDIO-CRE-mCherry-WPRE-hGH and AAV2/9-CaMKIIα-ArchT-EGFP into CLA and injected AAV2/R-CaMKIIα-FLP-WPRE-hGH into PL of Oprk1$^{lox/lox}$ mice (Fig. 6l). Validation of knockdown of KORs was confirmed by in situ hybridization (Fig. 6m, n). Kockdown of KORs in PL-projecting CLA glutamatergic neurons reversed optogenetic inhibition of CLA-PL circuit-induced pro-depressant effect, indicated by a significant increase in SI ratio in the SIT and sucrose preference in the SPT and decrease in immobility in the TST (Fig. 6o–q). Either overexpression or knockdown of KORs had no significant effect on locomotor activity (Fig. 6k, r). Taken together,

these data support that KORs play an essential role in CSDS-induced depressive-like behaviors by inhibiting excitatory transmission from the CLA to the PL.

## Discussion

This study reveals a molecular and circuitry mechanism underlying chronic stress induction of depression and highlights the importance of κ-opioid receptor-tuned excitatory inputs from the CLA to PL PV neurons for mediating CSDS-induced dysfunction of PL micronetwork and depressive-like behaviors (Fig. 7).

Dysfunction of cortical micronetworks through either increase of excitatory transmission or decrease of inhibitory transmission could give rise to the social and cognitive deficits observed in psychiatric diseases[7,8,53,54]. Substantial evidence has reported that dysfunction of cortical micronetworks mediated by impairment of GABAergic function is crucially involved in the aetiology of depression[10,55–57], although it is still unclear molecular and circuitry mechanisms underlying chronic stress dysfunction of cortical micronetwork. The PFC micronetwork may be particularly susceptible to dysfunction in response to

environmental stimuli due to its extensive reciprocally cortical and subcortical connectivity. This is supported by observations that stimulus related to psychiatric diseases alter PFC inhibitory transmission[53,58,59]. Although converging evidence supports an important role of the deficits in GABAergic transmission in the PFC in depressive disorders[10,47,56], it remains unknown that impairment of the strength of excitatory inputs from which subcortical region contributes to the deficits in GABAergic transmission in the PFC following chronic stress. Here, we demonstrate that loss of the CLA-PL excitatory drive to PL PV neurons results in chronic stress-induced deficits in GABAergic transmission and depressive-like behaviors.

The CLA is one of the least understood subcortical structures due to its complex shape and enclosed location[29]. The pervasive and reciprocal connectivity of the CLA with the PFC suggests a pivotal role of it in modulation of higher order brain functions. CLA has two important anatomical and functional features. Anatomically, its glutamatergic axons terminals most heavily distribute in deep layers of cortex[32,60–62]. Functionally, its synapses onto GABAergic neurons are stronger and faster than onto neighboring PNs and thus display much stronger FFI than many other structures that provide long-range FFI to cortical networks[32]. These two particular characteristics of the CLA make it a theoretically ideal subcortical region for modulating GABAergic neurons and control the inputs of the PNs in deep cortical layers through FFI. Deep cortical layers such as layer 5 is generally thought of as primarily an output layer and involved in the top-down control of other brain areas[63]. Considering CLA acting as a key relay node within complex functional neural networks, it may be particularly affected by neuropsychiatric diseases. Indeed, a significant decrease in the volume of the CLA is observed in patients with schizophrenia, autism and depression[64,65]. Niu with colleagues found that CLA was crucial for the control of stress-induced anxiety behaviors[66]. In this study we demonstrate that CLA exerts an importantly regulatory effect on PL micronetwork and plays crucial role in depressive-like behavior in mice.

PV interneurons are the largest population of interneurons in the cortex and present especially prominent in the deep layers[46,67]. PV neurons are thought to be critical for FFI[68,69], an efficient way to limit excess firing in neural networks and to synchronize spiking activity by setting network oscillations[70,71]. Previous studies have shown that stimulus-evoked, PV neurons-mediated gamma oscillations are important for refining information flow through cortical circuits[72], but high baseline gamma oscillations induced by elevated E/I balance may interfere with cortical function and contribute to psychiatric diseases[7,73]. Given that CLA axon terminals and PV interneurons display similar distributions in cortex and the CLA excitatory inputs to GABAergic interneurons are stronger than onto PNs, the CLA may perform a unique gating role among PFC inputs, functioning to balance stronger excitatory inputs from these other regions[74]. Impairment of excitatory inputs from the CLA to PV interneurons in the PFC could result in the micronetwork dysfunction in the PFC. In the present study, we demonstrated that PV neurons in the PL received direct monosynaptic inputs from the CLA and that PL PNs receive FFI from PV interneurons. We further demonstrated that impairment of the strength of excitatory inputs from the CLA to PV neurons within the PL contributed to the development of depressive-like behaviors.

KORs and its endogenous ligand dynorphin are powerful effectors of stress-induced psychiatric disorders such as anxiety and depression[52,75]. Accumulating evidence suggests that dynorphin/KOR signaling in the prefrontal cortex represents an important substrate for negative affects[76–81]. However, the circuitry crucial for this KOR-dependent regulation is unknown. The other major finding of this study is that we found that KORs were present in PL-projected CLA glutamatergic neurons, which negatively modulated excitatory transmission from the CLA to the PL. This notion is supported by the findings that activation of KORs with its agonist U50,488H decreased

optogenetic activation of the CLA-PL circuit-evoked EPSCs amplitude in PL PV neurons. The decreased amplitude was associated with an increase in PPR ratios of EPSCs, suggesting that KORs act presynaptically to decrease release of glutamate onto PV cells. These data are consistent with previous findings showing that mPFC KORs negatively regulate excitatory neurotransmission[78,81,82]. Furthermore, we observed that U50,488H inhibition of CLA-evoked oEPSCs and increased PPR was more profound in PL PV neurons than PNs. A possibility is that CLA neurons that express KOR target both PV interneurons and PNs in the PL, but PV neurons express higher level of KOR. Another possibility is that functional KORs are preferentially targeted to CLA synapses on PV, rather than PNs. Further studies will be necessary to determine the discrete role of dynorphin KOR signaling at projections into these different cell types for controlling the function of CLA-PL pathway and stress-induced depression. Our findings are consistent with that CLA neurons send biased excitatory input to GABAergic interneurons compared to PNs in the PFC[32]. Overall, our results reveal that KORs play an important inhibitory control over glutamatergic signaling within the PL by presynaptic mechanism and suggest that KORs may exert a key effect on PL micronetwork through modulation of excitatory drive from the CLA to PL PV neurons. We also demonstrated that chronic stress-evoked upregulation of KOR signaling was necessary and sufficient to develop chronic stress-induced depressive-like behaviors. Knockdown of KORs in CLA or infusion of KOR receptor antagonist nor-BNI into the PL counteracted optogenetic inhibition of the CLA-PL circuit-induced pro-depressant effect, whereas overexpression of KORs in CLA or local administration of KOR agonist U50,488H into PL reversed optogenetic activation of CLA-PL circuit-induced anti-depressant effect. The causal link between KOR signaling and PV neuronal function is supported by findings that local administration of GABA receptor antagonists into the PL inhibited optogenetic activation CLA-PL pathway-induced anti-depressant effect, as did intra-PL microinjection of KOR agonists, and that intra-PL administration of GABA receptor agonists reversed optogenetic inhibition of CLA-PL circuit-induced pro-depressant effect, as did intra-PL microinjection of the KOR antagonists.

The results of the current and other previous studies suggest that the recovery of E/I balance in cortical micronetwork may be a pivotal therapeutic strategy for depression associated with chronic stress. Ketamine, an antagonist of the NMDA receptor, produces rapid anti-depressant effect, although the mechanisms underlying such effect of ketamine are not fully elucidated. The counteraction of E/I imbalance may be an important cause, because it can rescue chronic stress-induced E/I imbalance either by suppression of excitation of glutamatergic neurons[83] or by potentiation of PV neuronal activity[47,55–57]. Substantial evidence also shows that ketamine predominately decreases the activity of GABAergic neurons and consequently increases pyramidal cell excitability, downstream of synaptic GABAergic disinhibition[84,85]. Given that dynorphin/KOR signaling modulates E/I balance in PL micronetwork and that E/I imbalance is linked to depression, our findings suggest that dynorphin/KOR signaling may be a potential target of the development of fast-acting antidepressant agents and the intervention for depression with social stress etiology.

In sum, our findings support for the hypothesis that depression is linked to disruption of homeostatic control of cortical micronetwork and unveil dynorphin/KOR signaling and CLA^Glu-PL^PV circuit as a signal and circuit-based mechanism by which chronic stress evokes depression through dysfunction of PL micronetwork via impairing excitatory inputs from the CLA to PL PV neurons.

## Methods
### Animals
Male C57BL/6 mice weighing 20–25 g were purchased from Shanghai Lingchang Biotechnology Co., Ltd. Retired CD-1 male breeders were purchased from Beijing Charles River Laboratory Animal Technology

Co., Ltd. PV-Cre mice (Jackson number: 008069) and Oprk1[lox/lox] (Jackson number: 030076) were used and genotyped according to the protocols provided by Jackson Laboratory. All mice were kept in a temperature (22.5 ± 1 °C) and humidity (50–60%) controlled room on a 12-h light/dark cycle with food and water ad libitum. Mice were housed in groups of up to five animals except for mice that underwent social defeat stress, which were housed singly to avoid fighting. All experiment protocols were approved by the Animal Care and Use Committee of Shanghai Institute of Materia Medica, Chinese Academy of Sciences.

## Drugs and chemicals
CNO (C0832), Baclofen (G013), Muscimol (G019), CNQX (C127), AP5 (A8054) and 4-AP (275875), Ibotenic acid (I2765) were purchased from Sigma-Aldrich. Zoletil was bought from Virbac. U50,488H (0495/25) was purchased from Tocris. nor-BNI (ab120078) was bought from Abcam. Bicuculline (abs812832) was bought from Absin. Saclofen (T4440) was bought from TOPSCIENCE. TTX (A0224) was purchased from Chengdu Must Bio-Technology Co., Ltd. The retrograde tracer Fluoro-Gold (sc-358883) was purchased from Santa Cruz Biotechnology.

## Viruses
AAV2/9-CaMKIIα-ChR2-EYFP, AAV2/9-CaMKIIα-ArchT-EGFP, AAV2/9-CaMKIIα-EGFP, AAV2/9-CaMKIIα-hM3Dq-mCherry, AAV2/9-CaMKIIα-hM4Di-mCherry, AAV2/9-CaMKIIα-mCherry, AAV2/9-EF1α-DIO-hM3Dq-mCherry, AAV2/9-EF1α-DIO-hM4Di-mCherry, AAV2/9-EF1α-DIO-mCherry, AAV2/9-EF1α-DIO-mCherry-RVG-WPRE-pA, AAV2/9-EF1α-DIO-mCherry-F2A-TVA-WPRE-hGH pA, AAV2/R-CaMKIIα-FLP-WPRE-hGH, AAV2/9-hSyn-fDIO-CRE-mCherry-WPRE-hGH, AAV2/R-CaMKIIα-Cre-WPRE-hGH and AAV2/9-CaMKIIα-GCaMP6s AAV2/9-DIO-GCaMP6s were purchased from BrainVTA Co., Ltd. The titers of above viral vectors were $2 \times 10^{12-13}$ vg/ml. RV-ENVA-ΔG-EGFP (viral titers: $2 \times 10^{8}$ IFU/ml) was purchased from BrainVTA Co., Ltd. AAV2/9-CaMKIIα-DIO-Oprk1-P2A-mCherry was purchased from OBIO Technology. All viral vectors were aliquoted and stored at −80 °C.

## Cannula implantation and microinjection
All stereotaxic surgeries were performed on a stereotaxic apparatus (RWD Life Science). Mice were anesthetized with sodium pentobarbital (70 mg/kg, i.p.) or Zoletil (75 mg/kg, s.c.). An incision was made on the mouse head to expose the skull surface. After scraping away the pericranium, burr holes were made on the skull. The 26-gauge guide cannulas were bilaterally implanted above the PL (AP: +1.98 mm; ML: ±0.3 mm; DV: −1.1 mm. AP, ML and DV denote anteroposterior, mediolateral and dorsoventral distance from the bregma respectively). The cannulas were anchored to the skull with stainless-steel screws and dental cement. Microinjections were made through a 33 gauge needle connected to a 10 μl Hamilton microsyringe mounted on a microinfusion pump (Harvard apparatus, USA). The needle extended 1 mm beyond the tips of the guide cannula to prevent blockage. For the activation or inhibition of the KOR, a total volume of 0.5 μl U50,488H (10 μg/μl in saline, 15 min before optogenetic procedures, given repeatedly) or 0.25 μl nor-BNI (10 μg/μl in saline, 24 h before optogenetic procedures, given once) was injected into the PL bilaterally. For the activation or inhibition of the GABAergic neurons in the PL, 0.5 μl GABA agonists mixture (baclofen, 0.2 μg/μl; muscimol, 0.2 μg/μl in saline, 15 min before optogenetic procedures, given once) or antagonist mixture (bicuculline, 0.02 μg/μl; saclofen, 0.9 μg/μl in saline, 15 min before optogenetic procedures, given repeatedly) was injected into the PL. The infusion rate of all drugs is 0.2 μl/min. After infusion, leave the needle in place for another 3 min to allow drug diffusion. The data was kept for analysis when the microinjection sites were in the right place.

## Viral injection and optical fiber implantation
Mice were anesthetized intraperitoneally with Zoletil (75 mg/kg). For chemogenetic viral delivery, microinjection was performed using the following stereotaxic coordinates for the CLA (AP: +1.1 mm; ML: ±2.5 mm; DV: −3.7 mm) (0.15 μl/side); PL (AP: +1.98 mm; ML: ±0.3 mm; DV: −2.1 mm) (0.2 μl/side) at a rate of 0.1 μl/min. After microinjection, the needle was left in place for 10 min to ensure virus diffusion and then carefully removed.

For optogenetic activation of the CLA-PL pathway, 0.15 μl of virus was microinjected into the unilateral CLA (AP: +1.1 mm; ML: +2.5 mm or −2.5 mm; DV: −3.7 mm) at a rate of 0.1 μl/min. An optical fiber (diameter, 200 μm; NA, 0.37, Inper) was implanted above the ipsilateral PL (AP: +1.98 mm; ML: +0.3 mm or −0.3 mm; DV: −1.6 mm). For optogenetic inhibition of the CLA-PL pathway, 0.15 μl of virus was microinjected into the CLA bilaterally (AP: +1.1 mm; ML: ±2.5 mm; DV: −3.7 mm) at a rate of 0.1 μl/min. Optical fibers were implanted above the PL bilaterally (AP: +1.98 mm; ML: ±0.6 mm; DV: −1.63 mm) at an angle of 14°. Tissue adhesive (Vetbond Tissue Adhesive; 3 M) was applied to the skull surface, and the optical fibers were anchored to the skull by two stainless-steel screws and dental cement. The optical fibers were connected to a laser source using an optical fiber sleeve (Newdoon).

For monosynaptic tracing with pseudotyped rabies virus, PV-Cre mice were microinjected with 0.1 μl of a viral cocktail (1:1) containing AAV-EF1α-DIO-mCherry-RVG and AAV-EF1α-DIO-mCherry-F2A-TVA into the PL (AP: +1.98 mm; ML: ±0.3 mm; DV: −2.1 mm). Three weeks later, 0.1 μl of the modified rabies virus RV-EvnA-ΔG-EGFP was microinjected into the same location. 5 days after the last injection, mice were killed and brain sections were collected for confocal imaging.

## Manipulation of KOR in the CLA[Glu]-PL circuit
For circuit over expression of KOR, 0.3 μl AAV-CaMKIIα-Cre (Retro) was injected into the PL, 0.5 μl of a viral cocktail (1:1) containing AAV-CaMKIIα-DIO-Oprk1-mCherry and AAV-CaMKIIα-ChR2-EYFP was injected into the CLA. For circuit knock out of KOR, 0.3 μl AAV-CaMKIIα-Flp (Retro) was injected into the PL, 0.5 μl of a viral cocktail (1:1) containing AAV-hSyn-fDIO-Cre-mCherry and AAV-CaMKIIα-ArchT-EGFP was injected into the CLA. Expressional validation of KOR was confirmed by in situ hybridization.

## Optical manipulation
The implanted fiber optics were connected to a fiber-coupled laser source (adjustable power: 0–100 mW; Anilab) through a rotatory joint patch cable (FC-M200 μm; NA 0.37; Anilab). Optical stimulation was controlled by stimulus generator software (AniOptover 2.0) via a computer. In optical activation tests, mice received 473 nm light stimulation (5 mW, 20 Hz, 5 ms pulses) in the period of social interaction test and tail suspension test. For the sucrose preference test, mice received 473 nm light stimulation (5 mW, 20 Hz, 5 ms pulses) for 6 cycles (3 min stimulation and then rested for 2 min without light stimulation) before the experiment. For ArchT-mediated inhibition, fibers were connected to a 532 nm green laser diode with constant light stimulation (5 mW, 9 s on, 1 s off) for 20 min during the SSDS procedures.

## CLA-PL disconnection
Mice were anesthetized by sodium pentobarbital (70 mg/kg, dissolved in saline, i.p.) and then secured to a stereotaxic apparatus (RWD). The head was leveled after exposing the surface of the skull and small holes were drilled above CLA on the left side and PL on the right side. Ibotenic acid (10 μg/μl, 0.1 μl) was separately injected into CLA (AP: +1.1 mm; ML: −2.5 mm; DV: −3.7 mm) and PL (AP: +2.0 mm; ML: +0.3 mm; DV: −2.1 mm) at a rate of 0.1 μl/min. The syringe was left in place for an additional 10 min to ensure drug diffusion. Behavioral tests were conducted 7 days after surgery. All mice were killed after the tests, and the brains were removed and sectioned for immunohistochemistry to verify neural lesion.

## Fiber photometry

Fiber photometry was used to record calcium signals from glutamatergic neurons or PV neurons in the CLA or PL using a two color multichannel optical fiber recording system (RWD Life Technology, R810). 0.2 µl of AAV-CaMKIIa-GCaMP6s virus or AAV-DIO-GCaMP6s was injected into the CLA or PL after mice were anesthetized with Zoletil, and an optical fiber (200 µm OD, 0.37 NA, Inper) was implanted into the CLA or PL. Three weeks after virus injection and fiber implantation, mice underwent CSDS model and SI test. $Ca^{2+}$ fluorescence was recorded during the SI test. Before recording, mice were allowed to rest and habituate for 3 min. Then, the $Ca^{2+}$ fluorescence was recorded during 3 min of baseline activity without an CD-1 mice, followed by 3 min of CD-1 exposure. 470 and 410 nm laser were used for GCaMP6s signal and autofluorescence measurement. The 410 nm channel served as the control channel and was subtracted from the GCaMP6s channel to eliminate autofluorescence, bleaching and motion effects. Light at the fiber tip ranged from 10 to 20 µW (410 nm) and 20 to 40 µW (470 nm) and was constant across trials over testing days. Change in fluorescence ($\Delta F/F = (F - F0)/F0$) during single social interaction behavior (−2 s to 6 s) or total social interaction (−180 s to 180 s) session was calculated and analyzed by the analysis software developed by RWD. F0 was defined as the mean baseline signals from −2 s to 0 s for single social interaction behavior or −180 s to 0 s for the total social interaction session.

Following recordings, mice were transcardially perfused with 4% paraformaldehyde (PFA) and processed to confirm viral expression and placement of optical fibers. The data was kept for analysis when the viruses and fibers were placed in the right sites.

## In vitro electrophysiology

The in vitro electrophysiology experiments were performed according to previous studies[51,86]. The mice brains were quickly removed under isoflurane anesthesia and immersed into 0 °C cutting solution (in mM: 234 Sucrose, 5 KCl, 1.25 $NaH_2PO_4$, 5 $MgSO_4$, 26 $NaHCO_3$, 25 Dextrose, 1 $CaCl_2$). Coronal slices including CLA or PL at 300 µm thickness were cut by a vibratome (VT1000S, Leica) and incubated at 25 °C in artificial cerebrospinal fluid (ACSF, in mM: 120 NaCl, 11 Dextrose, 2.5 KCl, 1.28 $MgSO_4$, 3.3 $CaCl_2$, 1 $NaH_2PO_4$, and 14.3 $NaHCO_3$) for 1 h. Then the slices were transferred into a recording chamber at 25 °C for in vitro electrophysiological recording. To test the membrane excitability of CLA neurons, episodic currents were injected under the current clamp configuration in 5 pA increments from 0 pA to depolarizing 150 pA. To verify the function of optogenetic and chemogenetic virus, neurons transfected with EGFP or mCherry were recorded in current clamp, and light stimulation or CNO (10 µM) was given. The patch pipette (5–10 Ω) was filled with the potassium-based recording solution (K-gluconate: 140 mM, NaCl: 5 mM, EGTA: 0.2 mM, Mg-ATP: 2 mM, HEPES: 10 mM, and 0.2% biocytin). To record spontaneous synaptic currents, a low divalent ion ACSF (in mM: 125 NaCl, 3.5 KCl, 1.25 $NaH_2PO_4$, 0.5 $MgCl_2$, 26 $NaHCO_3$, 25 Dextrose, and 1 $CaCl_2$) was used. Using pipettes filled with cesium-based recording solution (in mM: 100 $CsCH_3SO_3$, 20 KCl, 10 HEPES, 4 Mg-ATP, 0.3 Tris-GTP, 7 $Tris_2$-Phosphocreatine, and 3 QX-314), sEPSCs were recorded at a holding potential of −60 mV and sIPSCs were recorded at a holding potential of +10 mV. To verify the synaptic connectivity between CLA and PL, $PV^+$ neurons in the PL were identified by mCherry and recorded in voltage clamp. TTX (1 µM), 4-AP (100 µM), CNQX (40 µM) and AP5 (20 µM) were used to verify whether the light evoked currents were monosynaptic. To obtain the paired pulse ratio, evoked EPSCs were elicited by two focal light stimulations (pulse duration 2 ms) with an interval of 200 ms. KOR agonist U50,488H (1 µM) and antagonist nor-BNI (10 µM) were applied through perfusion. All the light stimulations were applied through Polygen 1000 (Mightex, Canada). All signals were amplified and recorded by a HEKA EPC10 amplifier (HEKA Instruments, Germany). Individual events were counted and analyzed with MiniAnalysis software (version 6.0.3).

## Behavioral procedures

**Social defeat stress paradigm.** Chronic social defeat stress (CSDS) and subthreshold social defeat stress (SSDS) paradigm were performed as described by previous studies with minor modification[41]. The CSDS paradigm was performed under the following procedures. Firstly, CD-1 aggressor mice were screened for three consecutive days with strict standard, which was that CD-1 mice must attack the C57BL/6 mouse for at least five consecutive sessions in 3 min, and the latency to initial aggression must be less than 60 s. Then, CD-1 mice were housed individually in cages that were separated into two compartments by perforated plastic separators. The C57 mice were placed into the home cages of CD-1 mice, and exposed to aggressive CD-1 for 10 min, during which time they were physically attacked by CD-1. After 10 min of physical contact, the mice were separated but maintained in sensory contact for 24 h using a perforated plexiglass partition. After 10 days of CSDS, animals were housed separately. Control animals were housed in pairs with one mouse per side of the perforated plexiglass partition, and they were never in physical or sensory contact with CD-1 mice before social interaction test. 24 h after the last social defeat stress, the behavioral procedures including social interaction test, tail suspension test, sucrose preference test and locomotor activity test were applied to evaluate depressive-like behaviors of mice.

For the SSDS paradigm, C57 mouse was placed into the home cage of a screened aggressor CD-1 for 10 min, during which time the C57 mouse was physically attacked. After 10 min of physical attack, they were physically separated with a perforated plexiglass partition, and the C57 mouse underwent 10 min of sensory stress. After 10 min of sensory stress, the intruder was returned to its home cage for 10 min, then it went through a second round of physical interaction and sensory stress in the home cage of a new CD-1 mouse. After two defeat bouts, C57 mouse was returned to its home cage and underwent behavioral tests after 24 h.

**Social interaction test.** Social interaction behavior, towards a novel CD-1 mouse, was measured in a two-stage social interaction test. In the first 3 min test (no target), the experimental C57 was allowed to freely explore a square shaped arena ($40 \times 40 \times 40$ cm³) containing an empty wire mesh cage ($10 \times 6.5 \times 20$ cm³) placed on one side of the arena. In the second 3 min test (target), the experimental C57 was reintroduced into the arena with an unfamiliar CD-1 mouse in the wire mesh cage. Video tracking software was used to measure the time of the experimental mouse spent in the "Interaction Zone" (a rectangular area (14 cm × 24 cm) around the enclosure that was applied to display the target CD-1 mice), "Corner Zone" (9 × 9 cm² zone projecting from both corner joints opposing the enclosure). The social interaction index was calculated as time spent in interaction zone with target/time spent in interaction zone without target.

## Tail suspension test

The tail suspension test was performed according to previous protocol[87]. Briefly, mice were suspended on their tails roughly 70 cm above the ground by adhesive tape (1 cm from the tail tip). The animals were video recorded for 6 min. The immobility time, defined as remaining motionless, was recorded for the last 4 min. Investigators were blinded to the treatment allocation when assessing behaviors.

## Forced swimming test

Animals were individually placed in a cylinder (25 cm height; 15 cm diameter) filled with water (23–25 °C) and swam for 6 min[87]. Water depth (15 cm height) was set to prevent animals from touching the bottom with their tails or hind limbs. Animal behaviors were video recorded from the side. The immobile time during the last 4 min was counted offline by an observer blinded to treatment allocation. Immobile time was defined when animals remained floating or motionless with only movements necessary for keeping balance in the water.

## Sucrose preference test

The sucrose preference test (SPT) was performed according to the previous studies[88]. Mice were initially habituated to two bottles of drinking water for two days, followed by 2 bottles of 1.5% sucrose solution for two days. After habituation, mice were deprived of water for 24 h. Then mice were given free access to a two-bottle choice of drinking water or 1.5% sucrose solution in the dark phase. Bottle positions were switched 1 h later (for 2 h test in the dark phase). Total consumption of each fluid was measured and sucrose preference score was calculated as: amount of sucrose consumed ×100%/(sucrose consumed + water consumed).

## Locomotor activity test

Mice were tested for their locomotor activity during 30 min in an open field arena ($40 \times 40 \times 40$ cm$^3$)[88]. Their motion tracks were recorded by the video camera, and analyzed with Shanghai Jiliang Software. Locomotor activity was evaluated as total distance traveled. For ChR2-stimulated experiments, locomotion was recorded throughout the arena for 10 min.

## Immunoblotting

After 10 days CSDS training, mice were decapitated under Zoletil anesthesia and the brains were rapidly removed on ice. Brain was cut into slices by knife blades held together with a 0.5 mm interstice, and the PL was dissected from the slice and stored at −80 °C. Brain tissues were homogenized in 60 µl RIPA buffer (50 mM Tris (pH 7.4), 150 mM NaCl, 1%Triton X-100, 1% sodium deoxycholate, 0.1% SDS, 2 mM sodium pyrophosphate, 25 mM β-glycerophosphate, 1 mM EDTA, 1 mM Na$_3$VO$_4$, 0.5 µg/ml leupeptin). A 30 µg protein aliquot from each sample was separated using SDS-polyacrylamide gel electrophoresis and transferred to a nitrocellulose membrane, which was then blocked with 5% BSA in 0.1% Tween 20 in TBS (TBST) for 2 h. The membranes were then incubated with primary antibodies against dynorphin A (1:300; Abcam, ab82509) and β-actin (1:3000, Cell Signaling Technology, 3700) overnight at 4 °C. After washing three times in TBST, the membranes were incubated with IRDye 800CW goat anti-mouse IgG secondary antibody (1:5000, LI-COR, 926-32210) and IRDye 680RD goat anti-rabbit IgG secondary antibody (1:5000, LI-COR, 926-68071) in 5% BSA for 2 h at room temperature. Quantitative analysis was performed with Image J software.

## Immunohistochemistry

Mice underwent behavioral analysis were deeply anesthetized with Zoletil and perfused transcardially with saline followed by 4% PFA in 0.1 M PBS. Brains were removed, post-fixed overnight in 4% PFA at 4 °C and transferred into 30% sucrose in 0.1 M PBS for dehydration. Coronal sections (30 µm) were cut on a cryostat (Leica, Germany) and stored in 0.1 M PBS. The sections were incubated in blocking buffer containing 1% bovine serum albumin (BSA) and 3% normal goat serum in 0.5% Triton X-100/PBS for 1.5 h at room temperature and then with primary antibodies in blocking buffer overnight at 4 °C. The primary antibodies used were: c-fos (1:1000, Synaptic system, 226003), PV (1:1000, Invitrogen, PA1-933) and Neurogranin (1:250, BioLegend, 845602). After three washes with PBS, sections were incubated with either Alexa Fluor 488- or Alexa Fluor 594-conjugated secondary antibodies at room temperature for 2 h. After another three washes in PBS, sections were incubated with DAPI (Beyotime, C1005) for 30 min and then washed three times. After rinsing, the sections were mounted and confocal images were captured with Olympus FV-1000, Zeiss LSM 710 or Leica TCS SPS CFSMP.

## KOR immunoprecipitation

Brain tissues of the PL were homogenized in 0.3 ml Buffer A (1% NP-40, 1% sodium deoxycholate, 0.1% SDS, 10 mM EDTA, 0.15 M NaCl, 25 mM Tris (pH 7.4)) with protease inhibitor mixture. 20 µg protein aliquot from each sample was separated using SDS-polyacrylamide gel electrophoresis, and the gel was incubated with GelCode Blue Safe Protein Stain (Thermo Scientific, 24594) for 2 h. Immunoprecipitations were performed using the KOR primary antibody PA847 (1 µg/µl, provided by Liu-Chen LY Lab) and Pansorbin (Sigma-Aldrich, 507858). Immunoprecipitated proteins were resolved by SDS-PAGE and transferred to PVDF membrane for immunoblotting. The membrane was blotted with PA847 (1:10000) overnight at 4°C and incubated with secondary antibody (HRP-anti-rabbit IgG light chain, 1:10000, Jackson ImmunoResearch, 211-032-171) at room temperature for 2 h. Quantitative analysis was performed with Image J software.

## In situ hybridization

Mice were deeply anesthetized with Zoletil and perfused with DEPC-PBS and 4% PFA. Brains were removed and post fixed overnight in 4% PFA at 4 °C. After being dehydrated in 30% DEPC-sucrose, coronal sections (12 µm) were made on a cryostat microtome and then mounted on Superfrost Plus microscope slides (Fisher brand, 12-550-15), dried at room temperature and stored at −80°C. The in situ hybridization procedures were performed according to the manufacturer's instructions with minor modifications of RNAScope Multiplex Flourescent Reagent Kit V2 (Advanced Cell Diagnostics, 323100). Briefly, slides were washed once in DEPC-water and 3 times in 100% ethanol and then totally dried at 60°C for 2 h. After cooling at room temperature, slides were pretreated with hydrogen peroxide for 20 min and then boiled in target retrieval reagents for 7 min at 100 °C and rinsed 3 times with DEPC-water and 3 times with 100% ethanol. Sections were then incubated with protease III for 30 min and rinsed 3 times with DEPC-water before incubating with probes for vGlut1-C1 (Slc17a7, ACD, 416631-C1), mCherry-C2 (ACD, 431201-C2), Oprk1-C2 (ACD, 316111-C2), GAD65-C3 (ACD, 439371-C3), mCherry-C3 (ACD, 431201-C3) for 2 h. All incubation steps were performed at 40 °C using a HybEz oven. After probe incubation, slides were rinsed 3 times with wash buffer reagent followed by Amplification step I (30 min), Amplification step II (30 min) and Amplification step III (15 min) with two washes between steps. Sections were then rinsed 2 times with wash buffer reagent before incubating with HRP-C1 for 15 min followed by two washes. Sections were incubated with C1-probe associated Opal 520 (1:1000) for 30 min followed by two washes and a subsequent incubation in HRP blocker solution for 15 min followed by two washes. The HRP-C2 and HRP-C3 procedures were the same as the HRP-C1, except for incubating with C2-probe associated Opal 570 (1:1500) and C3-probe associated Opal 690 (1:750) respectively. Sections were incubated with DAPI for 1 min and treated with the Prolong Gold Antifade Reagent (Invitrogen P36930) before carefully mounted with cover glass.

## Quantification and statistical analyses

All data were presented as means ± s.e.m. Number of experimental replicates (n) was indicated in the figure legends and referred to the number of experimental subjects independently treated in each experimental condition. Statistical comparisons were performed using GraphPad Prism (version 8.0) with appropriate methods as indicated in the figure legends. Single-variable differences were analyzed by two-tailed and unpaired Student's t tests. Grouped differences were analyzed with One- or Two-way analysis of variance (ANOVA) followed by Bonferroni post-hoc test. Statistical significance was set at $^{*}P < 0.05$, $^{**}P < 0.01$, $^{***}P < 0.001$, $^{****}P < 0.0001$.

## Reporting summary

Further information on research design is available in the Nature Portfolio Reporting Summary linked to this article.

## Data availability

All data needed to evaluate the conclusions in the paper are presented in the results and/or supplementary materials. Any additional

information is available from the corresponding author. Source data are provided with this paper.

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

## Acknowledgements

The authors would like to thank Dr. Lee-Yuan Liu-Chen and Chongguang Chen, Temple University Lewis Katz School of Medicine, for providing KOR antibody PA847 with a detailed detecting protocol. This research was supported by the Major Project of the Science and Technology Innovation 2030 of China (STI2030-Major Projects 2021ZD0202900 to J.-G.L., 2021ZD0203500 to Y.-J.W.), from the National Natural Science Foundation of China (82030112 to J.-G.L., 81773710 to Y.-J.W., 82273904 to G.-Y.Z.), from Science and Technology Commission of Shanghai Municipality (20ZR1468200, 23ZR1474900 to Y.-J.W.) and from Shenzhen-Hong Kong Institute of Brain Science—Shenzhen Fundamental Research Institutions (NYKFKT2019015 to J.-G.L.).

## Author contributions

J.-G.L., Y.-J.W., L.X. and Z.C. designed the experiments. Y.-J.W., G.-Y.Z., C.X., X.-P.L., X.-L.S., S.-Y.Y., X.-S.X., X.Q. and Y.C. performed the experiments and statistical analysis with the assistance of K.J., Q.-X.Z., J.-Y.Y. and Y.W. The manuscript was written by Y.-J.W., G.-Y.Z., X.-L.S. and J.-G.L. and was revised by L.X., Z.C. and J.-G.L.

## Competing interests

The authors declare no competing interests.
