## [Peer Review File · Nature Communications]

The claustrum-prelimbic cortex circuit through dynorphin/ κ -opioid receptor signaling underlies depression associated with social stress etiologyREVIEWER COMMENTS

Reviewer #1 (Remarks to the Author):

Comments to the authors (manuscript # 428001)

The claustrum (CLA) is known to be connected with the prefrontal cortex, and both are crucial for controlling stress-induced anxiety-related behaviors. The CLA also has the highest level of the kappa opioid receptor (KOR) in the brain. However, whether this circuit connection is associated with stress etiology of depression and how KOR signaling is involved in this process remains to be determined. In this study, Wang et al. demonstrate a novel pathway by which claustrum neurons that project to the prelimbic region of the prefrontal cortex play a crucial role in the development of depressive-like behaviors after chronic social defeat stress in mice. In addition to chronic social defeat stress, the authors show that mice manipulated with subthreshold social defeat stress do not display depressive behaviors. They further employ whole-cell electrophysiological recordings and optogenetics to demonstrate that CLA excitatory cells make monosynaptic connections with both pyramidal neurons and PV interneurons in the mPFC in wild-type and PV-cre mice. Using c-fos expression and fiber photometry, they find that CLA glutamatergic neurons and PV interneurons in PL decrease in activity, but pyramidal neurons in PL increase in activity after CSDS. The authors also use chemogenetics, optogenetics, and the combination of the two to bidirectionally modulate inputs from the CLA onto both pyramidal and PV neurons in PL. They find that chemogenetic activation of PV interneurons reverses the decrease in depressive behaviors induced by photogenetic CLA to PL pathway inhibition. Importantly, the authors use circuit-specific viral kappa opioid receptor knockdown and overexpression to highlight the importance of opioid signaling to the observed behavioral effect.

This study presents a series of high-quality data that are generally convincing, and these findings provide a novel insight into the circuit mechanism of how stress may induce depression-like behavior and potential intervention by targeting the CLA-PL circuit and KOR signaling. However, there is a concern that the authors manipulate the circuit before the SSDS paradigm but never before the CSDS. The circuit in CSDS experiments is manipulated at the point of expression of depressive-like behavior. Thus, it is not clear if this circuit is important in the acquisition of depressive state, expression, of depressive behaviors, or both. There are also other concerns that need to be addressed.

Major issues

1. Co-localization of GAD65 and mCherry in Supplementary Figure 2 is concerning. It is not possible to ascertain that the CNO-stimulated population in the experiment is glutamatergic. The CLA contains 85% glutamatergic and 10-15% GABA-ergic neurons, so the chosen promoter is not selective for glutamatergic neurons and will infect the cells unbiasedly.
2. In Supplementary Figure 4D, the amplitude of EPSC on sample recording seems to be lower in the CSDS condition compared to the control. The summarized amplitude is shown to be not significantly altered in Sup Fig4 F. Although this might be driven by one outlier cell, it is important to show a typical example for the data. There is also a trend of decreases in the EPSC amplitude. Additionally, given the significantly reduced EPSC amplitude by KOR manipulation, the evidence of presynaptic mechanism based on altered PPR only is weak.
3. It is not possible to claim like the authors do that they demonstrated that "chronic stress-evoked upregulation of KOR signaling was necessary and sufficient to develop chronic stress-induced depressive-like behaviors." neither the KOR overexpression model nor the KOR knockdown showed different responses to the CSDS or SSDS paradigms (Fig 6c and Fig 6 I). The behavioral differences in response to optogenetic stimulation or inhibition of the pathway may be accounted for by KOR receptors being present on multiple cell types and incoming projections in the PFC.

4. The authors repeatedly used "data not shown" for some crucial or supporting data. It is important to show these data in the supplemental information. For example, in lines 324-331, chemogenetic inhibition or activation of PV cells affects the expression of c-Fos expression in the PL. Is the c-Fos expression in the pyramidal neurons only or both pyramidal cells and PV/GAD65 interneurons? It is easy to imagine an altered c-fos expression by inhibiting PV, but it is hard to believe how activation of PV cells may affect c-fos expression in the pyramidal neurons when their activity is decreased? These findings are important for data interpretation.

5. In the experiment described in Figure 6, the kappa opioid receptor overexpression and knockdown viruses are DIO- and fDIO driven, dependent on Cre or FLP injected into the PL. The optogenetic-carrying viruses injected into CLA are not driven by the CRE or FLP, and so are not specific for CLA to PL projection.

Minor issues

1. In the fiber photometry experiment described in Figure 2 the duration of the experiments is different between the PV-cre animals (Fig2P-R) and the other two groups. It is not clear why the data is presented in this way and not uniformly.

2. In Figure 4C it seems that not all AAV-Helper -mCherry labeled neurons co-localize with PV marker.

3. In addition to demonstrating the effect of KOR agonist U50,488H in Supplementary Figure 7, it would be desirable to show the effect of KOR antagonist nor-BNI on both oEPSP amplitude and PPR.

4. KOR staining for KOR-knockout mice in Figure 6 still shows robust KOR expression. Quantifying how this expression compares to the wild type would be beneficial.

5. Beta-actin bands in Supplementary Figure 8A do not appear uniform in thickness.

6. Niu et al. Sci Adv. 2022 Mar; 8(11): eabi6375 paper may need to be cited.

Reviewer #2 (Remarks to the Author):

The present study by Wang et al test the hypothesis that the CLA to PL pathway drives resilience against stressors by engaging feedforward inhibitory mechanisms that decrease activity of the PL neurons via activation of PV neurons. They also show that this process is disrupted by increased dynorphin / KOR signaling at CLA to PV synapses. Overall, the study is thorough and rigorous and will be of interest to a broad audience. The authors do an excellent job of testing their hypothesis with converging approaches and clever studies. There is plenty of data the authors refer that is not shown but the description of those data sound like they strongly enhance the conclusions of the present study. One challenge that is faced in the present study is there is enough work completed that this could ultimately comprise two different papers. The authors have a lot of evidence that is not truly supplemental or that they don't show since the paper is so dense. Lots of information, especially with regards to the dynorphin / KOR, work gets lost due to figure constraints. One consideration may be to split the study to two manuscripts (a CLA to PFC one and another on dynorphin / KOR regulation) and focus on the CLA to PFC in the present study and present the entirety of their work and address some of the relatively minor issues that are associated with this portion of the study. Otherwise the authors have done a great job here. Below are comments/suggestions that would further increase the strength of the paper.

- The authors lead into the results by stating that they have data demonstrating that ibotenic acid

lesions of CLA leads to depressive like behavior. These data should be shown, at the very least in supplemental figures, or remove the statement. This applies to the other examples of data that is not shown that I highlight below.

- Likewise the disconnection experiment is also exciting. The inclusion of those results would nicely increase the rigor of the study.
- In lines 129-130, the authors state that CNO was administered 24 hrs after the SDSS but in Fig 1A the diagram shows that CNO was administered prior to SDSS. From the methods, it is unclear when the CNO was administered. These inconsistencies should be clarified.
- For the chemogenetic experiments in Fig. S2e-i, did the authors administer CNO repeatedly before each test? Or was it only one injection similar to the experiments in Fig 1a-g?
- The figures were of low resolution in the provided manuscript so some of the data were difficult to judge. For example, for the immunohistochemistry results in Fig. 2, the electrophysiology EPSC data in Fig S4.
- In the electrophysiology data in Fig. S4 the noise in the representative traces is very high and the amplitude of events are very large. Is it possible that the authors recordings biasing towards only detecting large events given the high levels of noise? This may not change the overall interpretation but is a caveat that should be discussed surrounding this finding.
- The authors report that activity in CLA and PL PV neurons is decreased with fiber photometry experiments but they are comparing activity during different behaviors (social interaction vs tail suspension). Thus, it is difficult to interpret these data. Performing the same behavior in PV neurons would be necessary to be able to draw conclusions across the populations in fiber photometry experiments.
- The authors should also show the cFos data with chemogenetic manipulation of PV neurons in lines 324-330. The description of these data sound like they strengthen the authors conclusions but without seeing the data it is hard to ascertain how much weight is placed on the text.
- The authors propose that CLA targets deep layers and this may recruit PV neurons in deep layers. Analysis of the of cFos across layers from the authors existing data set may be useful in strengthening this statement and correlating it to cfos activation of pyramidal neurons in a layer-specific manner.
- The authors interpretation that KOR cells preferentially project to PV neurons should be tempered as alternative explanations are possible. The distinct possibilities should be discussed.
- The authors should include the nor-BNI data on EPSCs in PV neurons they state is not shown.
- For the microinjection procedures the authors should directly state whether microinjections were performed repeatedly in the same mice.
- Quantification of the KOR overexpression images would strengthen establishment of this approach.
- The results obtained from immunoprecipitation of KOR are interesting but antibodies for KORs, and GPCRs in general, are notoriously non-specific . Appropriate validation of the KOR antibody would be necessary or at the very least discussion of the caveat of not validating this novel antibody would be necessary.
- The authors can do a better job of covering the literature in the following ways. I suggest the authors carefully vet the way they are citing literature and be sure to incorporate papers that also highlight the discrepancies in the field. Most importantly, the authors ignore the literature that exists in on the PFC Dyn system and cite it's actions elsewhere in the brain.
 - o There are several studies that have studied the dynorphin / KOR system in PFC showing that PFC KOR plays a role in negative affect that the authors need to cite and discuss in the context of the present studies, including Abraham et al 2022, Wall and Messier 2015, Tejada et al 2013, 2015, Bals-Kubik et al 1993, Fassini et al 2015
 - o The authors should cite work from Tejada et al 2013; 2015 and Yarur-Castillo et al 2022 showing that KOR regulates excitatory synapses in the PFC.
 - o The authors also selectively discuss the literature on interneurons in PFC selectively or in inaccurately. Increased
 - o As another example, studies have demonstrated that ketamine and other NMDAR antagonists inhibit PV interneuron activity, counter to how the authors discuss the actions of ketamine in the context of their proposed model. When the authors do mention the NMDAR antagonism mechanisms of ketamine, they do not refer to this vast literature.

o Another example is citation of work on SST neurons by Fuchs et al in their discussion of potentiation of PV neurons

Minor comments:

- Line 110-111: the authors state that mice with SSDS displayed vulnerability to subsequent stress in a way that leads the reader to believe this is something they are showing in the present manuscript. Please update the language accordingly to say this is something established in the literature.

Reviewer #3 (Remarks to the Author):

In this study, Wang and colleagues investigate the molecular and circuit mechanisms of stress-induced disruption of cortical micro-network. The authors showed that the CSDS decreased the output of CLA-PL glutamatergic circuit. Optogenetic manipulation of this circuit bidirectionally regulated the depression-related behaviors induced by CSDS. Their further experiments provide the molecular mechanism that dynorphin/KOR signaling in CLA-PL circuit was involved in the depression-related behaviors. Overall, this is a very interesting and well-completed study. The experiments used elegant viral approaches, are well designed and executed to explore the circuit synaptic connection and function. There are some minor comments that should be addressed before publication.

The findings on dynorphin are an important part for this study. Some dynorphin results should be moved from supplementary figure 8 to main figure 6.

In Figure 3, it is confusing that different colors are used for groups. It would be better to use one color for all Chr2 groups, and another for all ArchT groups.

The labels could be better in Figure 5. It looks like CSDS did not work. Actually, all black bars in the bar graph should be Chr2 groups or ArchT groups, and other colored bars should be Chr2 + CNO groups or ArchT + CNO groups.

In the figure legend for Figure 2, the label (-180 s - 180 s) may change to (-180 s to +180 S); similarly, (-60 s - 380 s) to (-60 s to + 380). In addition, it is unclear why the recording ranges are different between the social interaction (panel 2d and 2j) and tail suspension test (panel 2p). Related information should be provided.

In the abstract, "being critical for 35 depression in mice" should be rephrased. It would be better to use depression-related behaviors in mice.

Also in the abstract, "and possibly intervening depression by targeting CLA-PL circuit" needs to be rewritten. The sentence is not complete.

Move CLA label to the location of LCA in Figure 1b and 1i.

More information about the neurogranin should be included, to explain why the authors used neurogranin antibody to label those neurons.

In the figure legend of Figure 3, "regulate mice depression-like behaviors" may change to "regulate depression-like behaviors of mice".

Responses to reviewer's comments

We greatly appreciate reviewers' efforts in reviewing our manuscript. These critical comments raised by reviewers are very important for improving our manuscript. We have addressed reviewers' all concerns either by new experiments or explanations.

REVIEWER COMMENTS

Reviewer #1 (Remarks to the Author):

Comments to the authors (manuscript # 428001)

The claustrum (CLA) is known to be connected with the prefrontal cortex, and both are crucial for controlling stress-induced anxiety-related behaviors. The CLA also has the highest level of the kappa opioid receptor (KOR) in the brain. However, whether this circuit connection is associated with stress etiology of depression and how KOR signaling is involved in this process remains to be determined. In this study, Wang et al. demonstrate a novel pathway by which claustrum neurons that project to the prelimbic region of the prefrontal cortex play a crucial role in the development of depressive-like behaviors after chronic social defeat stress in mice. In addition to chronic social defeat stress, the authors show that mice manipulated with subthreshold social defeat stress do not display depressive behaviors. They further employ whole-cell electrophysiological recordings and optogenetics to demonstrate that CLA excitatory cells make monosynaptic connections with both pyramidal neurons and PV interneurons in the mPFC in wild-type and PV-cre mice. Using c-fos expression and fiber photometry, they find that CLA glutamatergic neurons and PV interneurons in PL decrease in activity, but pyramidal neurons in PL increase in activity after CSDS. The authors also use chemogenetics, optogenetics, and the combination of the two to bidirectionally modulate inputs from the CLA onto both pyramidal and PV neurons in PL. They find that chemogenetic activation of PV interneurons reverses the decrease in depressive behaviors induced by photogenetic CLA to PL pathway inhibition. Importantly, the authors use circuit-specific viral kappa opioid receptor knockdown and overexpression to highlight the importance of opioid signaling to the observed behavioral effect.

This study presents a series of high-quality data that are generally convincing, and these findings provide a novel insight into the circuit mechanism of how stress may induce depression-like behavior and potential intervention by targeting the CLA-PL circuit and KOR signaling. However, there is a concern that the authors manipulate the circuit before the SSDS paradigm but never before the CSDS. The circuit in CSDS experiments is manipulated at the point of expression of depressive-like behavior. Thus, it is not clear if this circuit is important in the acquisition of depressive state, expression, of depressive behaviors, or both. There are also other concerns that need to be addressed.

Response: We thank reviewer for the positive and constructive comments, which are very important for improving the quality of our work. In the present work, we manipulated PL-projecting glutamatergic neurons before SSDS, and found that optogenetic inhibiting of the CLA-PL circuit promoted susceptibility to SSDS. In the CSDS paradigm, we manipulated PL-projecting glutamatergic neurons after CSDS, and found that optogenetic exciting of the CLA-PL circuit inhibited CSDS-induced depressive-like behaviors. These data confirmed the functional importance of the CLA-PL circuit and suggested that the CLA-PL circuit might be implicated in both the acquisition and expression phases of depressive behaviors.

Major issues

1. *Co-localization of GAD65 and mCherry in Supplementary Figure 2 is concerning. It is not possible to ascertain that the CNO-stimulated population in the experiment is glutamatergic. The CLA contains 85% glutamatergic and 10-15% GABA-ergic neurons, so the chosen promoter is not selective for glutamatergic neurons and will infect the cells unbiasedly.*

Response: In this study, the chosen promoter is CaMKII α , which is a known marker of excitatory neurons. Previous studies have shown that roughly 85% of claustral projection neurons are glutamatergic with expressing gene encoding vGlut2 (Mathur, 2014; Gomez-Urquijo et al., 2000). Ideally AAV expressing mCherry-tagged Gi or Gq DREADD under the control of CaMKII α should be restricted to glutamatergic neurons, however, we found that 85% colocalization of mCherry and vGlut 1, 9% colocalization of mCherry and GAD65. It could be due to leaky virus expression or off-target infection. Several studies have found that under many conditions in both responder mouse lines and with helper viruses, there is the potential for low levels of “leak” expression (Seidler et al., 2008; Wall et al., 2010; Miyamichi et al., 2013). To better address reviewer’s concern, we conducted additional experiments, and we still found that virus containing a CaMKII α promoter predominantly expressed in glutamatergic neurons in the CLA, with most (90.36%) mCherry⁺ neurons co-expressed with vGlut1⁺ mRNA, a small fraction (9.64%) expressing vGlut1⁻.

2. *In Supplementary Figure 4D, the amplitude of EPSC on sample recording seems to be lower in the CSDS condition compared to the control. The summarized amplitude is shown to be not significantly altered in Sup Fig4 F. Although this might be driven by one outlier cell, it is important to show a typical example for the data. There is also a trend of decreases in the EPSC amplitude. Additionally, given the significantly reduced EPSC amplitude by KOR manipulation, the evidence of presynaptic mechanism based on altered PPR only is weak.*

Response: To address reviewer’s concern, we have conducted additional experiments to redetecting the sEPSC of PV neurons in the PL of CSDS mice. We still found that the frequency of sEPSCs was decreased, but the amplitude of sEPSCs was not altered (Suppl Fig 5 d-f in the revised manuscript), which was consistent with previous data (Fig 4 d-f in the original manuscript). These findings, together with our in situ hybridization data showing that KOR and vGlut1 overlapped in the CLA and in vitro whole-cell electrophysiology recording data showing that application of KOR agonist U50,488H reduced the amplitude of oEPSCs and simultaneously increased the paired pulse ratio (PPR) in both PV and PNs in the PL (Suppl Fig 10), suggest that KORs negatively modulate synaptic transmission perhaps by inhibiting the probability of presynaptic glutamate release.

3. *It is not possible to claim like the authors do that they demonstrated that “chronic stress-evoked upregulation of KOR signaling was necessary and sufficient to develop chronic stress-induced depressive-like behaviors.” neither the KOR overexpression model nor the KOR knockdown showed different responses to the CSDS or SSDS paradigms (Fig 6c and Fig 6 I). The behavioral differences in response to optogenetic stimulation or inhibition of the pathway may be accounted for by KOR receptors being present on multiple cell types and incoming projections in the PFC.*

Response: In this study, we find that KORs are present in PL-projected CLA glutamatergic neurons. Chronic stress results in endogenous dynorphin release and increase KOR signaling in the PL, which impairs excitatory transmission from the CLA to PL PV interneurons, leading to high excitability of PL PNs and depressive-like behaviors via disinhibiting PNs. To confirm functional importance of KORs in modulation of depressive-like behaviors by regulating CLA afferents to the PL, we

manipulated dynorphin/KORs signaling by activation and blockade of KORs with its agonist and antagonist and by alterations of KOR levels with overexpression and knockout during optogenetic activation and inhibition of the CLA-PL circuit. We found that knockdown of KORs in CLA or infusion of KOR receptor antagonist nor-BNI into the PL counteracted optogenetic inhibition of the CLA-PL circuit-induced pro-depressant effect, whereas overexpression of KORs in CLA or local administration of KOR agonist U50,488H into PL reversed optogenetic activation of CLA-PL circuit-induced anti-depressant effect. For the detail, please see Fig 6e-k, l-r, Suppl Fig 11 d-g, h-k.

We manipulated KOR levels in PL-projecting CLA neurons by the strategies of circuitry specific overexpression and knockdown. KORs manipulations were limited in PL-projecting CLA glutamatergic neurons. For overexpression of KORs, we injected AAV2/9-CaMKII α -DIO-Oprk1-P2A-mCherry and AAV2/9-CaMKII α -Chr2-EYFP into CLA and AAV2/R-CaMKII α -Cre-WPRE-hGH into PL, whereas for knockdown of KOR, we injected AAV2/9-hSyn-fDIO-CRE-mCherry-WPRE-hGH and AAV2/9-CaMKII α -ArchT-EGFP into CLA and injected AAV2/R-CaMKII α -FLP-WPRE-hGH into PL of Oprk1^{lox/lox} mice. For the detail, please see Fig 6e, l.

4. The authors repeatedly used “data not shown” for some crucial or supporting data. It is important to show these data in the supplemental information. For example, in lines 324-331, chemogenetic inhibition or activation of PV cells affects the expression of c-Fos expression in the PL. Is the c-Fos expression in the pyramidal neurons only or both pyramidal cells and PV/GAD65 interneurons? It is easy to imagine an altered c-fos expression by inhibiting PV, but it is hard to believe how activation of PV cells may affect c-fos expression in the pyramidal neurons when their activity is decreased? These findings are important for data interpretation.

Response: We thank reviewer for the constructive suggestion. We actually counted the proportion of c-fos positive cells among mCherry-labelled PV interneurons. The results confirmed the effects of inhibitory hM4Di and excitatory hM3Dq on PL PV neurons. The c-fos data has been added in revised manuscript as Suppl Fig. 9. We also have provided the ibotenic acid lesion data and disconnection data as suppl Fig. 1, 7.

5. In the experiment described in Figure 6, the kappa opioid receptor overexpression and knockdown viruses are DIO- and fDIO driven, dependent on Cre or FLP injected into the PL. The optogenetic-carrying viruses injected into CLA are not driven by the CRE or FLP, and so are not specific for CLA to PL projection.

Response: Optogenetic manipulation was specific for CLA-PL projection, since we selectively activate or inhibit the CLA-PL circuit by injecting AAV-CaMKII α -Chr2-EYFP or AAV-CaMKII α -ArchT-EGFP into the CLA, and implanting the optical fibers over the PL.

Minor issues

1. In the fiber photometry experiment described in Figure 2 the duration of the experiments is different between the PV-cre animals (Fig2P-R) and the other two groups. It is not clear why the data is presented in this way and not uniformly.

Response: The data has been presented uniformly as reviewer suggested. Please see revised Fig 2 p-r.

2. In Figure 4C it seems that not all AAV-Helper -mCherry labeled neurons co-localize with PV

marker.

Response: Although monosynaptic circuit tracing with RVdG approach is widely used for sophisticated circuit-tracing studies throughout the central nervous system, the important limitation is incomplete labeling. The limiting factors include the time available for trans-synaptic spread before starter cells die, the initial numbers of RVdG particles entering the starter cells, and levels of G expression in starter cells (Callaway and Luo, 2015). The leaky TVA/tTA expression and concomitant off-target RV infection are also possible, and the helper virus concentrations may critically affect the efficiency of transsynaptic spread and nonspecific labeling (Lavin et al., 2020). In addition, as we mentioned in concern 1, ideally Cre-dependent gene expression should be absolutely restricted to Cre expressing neurons. However, under many conditions in both responder mouse lines and with helper viruses there is the potential for low levels of “leak” expression (Seidler et al., 2008; Wall et al., 2010; Miyamichi et al., 2013).

3. *In addition to demonstrating the effect of KOR agonist U50,488H in Supplementary Figure 7, it would be desirable to show the effect of KOR antagonist nor-BNI on both oEPSP amplitude and PPR.*

Response: We thank reviewer for the constructive suggestion. The norBNI data has been added as reviewer suggested. Please see revised Suppl Fig 10 i-k.

4. *KOR staining for KOR-knockout mice in Figure 6 still shows robust KOR expression. Quantifying how this expression compares to the wild type would be beneficial.*

Response: Yes, KOR-knockout group can still see certain KOR expression in Fig 6m, but compared with the control group, KOR expression level was significantly decreased. We have quantified situ hybridization data and added as Fig 6n. We also have changed “knockout” to “knockdown” in the revised version of manuscript.

5. *Beta-actin bands in Supplementary Figure 8A do not appear uniform in thickness.*

Response: Yes, the thickness of actin was not very even. However, by quantifying dynorphin A/actin, CSDS group shows higher ratio compared to the control group.

6. *Niu et al. Sci Adv. 2022 Mar; 8(11): eabi6375 paper may need to be cited.*

Response: This reference has been added as reviewer suggested. Please see page 17, the first sentence, line 8-10.

Reference:

Brian N. Mathur, B.N. (2014) The claustrum in review. *Front Syst Neurosci* 8, 48.

Callaway, E.M., and Luo, L. (2015). Monosynaptic Circuit Tracing with Glycoprotein-Deleted Rabies Viruses. *The Journal of Neuroscience* 35, 8979

Gomez-Urquijo S.M., Gutierrez-Ibarluzea I., Bueno-Lopez J.L., Reblet C. (2000). Percentage incidence of gamma-aminobutyric acid neurons in the claustrum of the rabbit and comparison with the cortex and putamen. *Neurosci Lett* 282:177–180.

Jackson J., Karnani, M.M., Zemelman, B.V., Burdakov, D., Lee, A.K. (2018) Inhibitory Control of Prefrontal Cortex by the Claustrum. *Neuron* 99, 1029–1039

Lavin, T.K., Jin, L., Lea, N.E., and Wickersham, I.R. (2020). Monosynaptic Tracing Success Depends Critically on Helper Virus Concentrations. *Front Synaptic Neurosci* 12, 492637.

Miyamichi, K., Shlomei-Fuchs, Y., Shu, M., Weissbourd, B.C., Luo, L., and Mizrahi, A. (2013). Dissecting Local Circuits: Parvalbumin Interneurons Underlie Broad Feedback Control of Olfactory Bulb Output. *Neuron* 80, 1232–1245.

Seidler, B., Schmidt, A., Mayr, U., Nakhai, H., Schmid, R.M., Schneider, G., and Saur, D. (2008). A Cre-loxP-based mouse model for conditional somatic gene expression and knockdown in vivo by using avian retroviral vectors. *Proc Natl Acad Sci U S A* 105, 10137–10142.

Wall, N.R., Wickersham, I.R., Cetin, A., De La Parra, M., and Callaway, E.M. (2010). Monosynaptic circuit tracing in vivo through Cre-dependent targeting and complementation of modified rabies virus. *Proc Natl Acad Sci U S A* 107, 21848–21853.

Reviewer #2 (Remarks to the Author):

The present study by Wang et al test the hypothesis that the CLA to PL pathway drives resilience against stressors by engaging feedforward inhibitory mechanisms that decrease activity of the PL neurons via activation of PV neurons. They also show that this process is disrupted by increased dynorphin / KOR signaling at CLA to PV synapses. Overall, the study is thorough and rigorous and will be of interest to a broad audience. The authors do an excellent job of testing their hypothesis with converging approaches and clever studies. There is plenty of data the authors refer that is not shown but the description of those data sound like they strongly enhance the conclusions of the present study. One challenge that is faced in the present study is there is enough work completed that this could ultimately comprise two different papers. The authors have a lot of evidence that is not truly supplemental or that they don't show since the paper is so dense. Lots of information, especially with regards to the dynorphin / KOR, work gets lost due to figure constraints. One consideration may be to split the study to two manuscripts (a CLA to PFC one and another on dynorphin / KOR regulation) and focus on the CLA to PFC in the present study and present the entirety of their work and address some of the relatively minor issues that are associated with this portion of the study. Otherwise the authors have done a great job here. Below are comments/suggestions that would further increase the strength of the paper.

Response: We thank reviewer for the positive comments and the constructive suggestions, which are very important for improving the quality of our work. This work originally started with the anatomy and the function of CLA-PL pathway in stress-induced depression, but during the work, we found an essential role of KORs within this circuit. Thus, these data were combined and we believe that it may provide a more comprehensive understanding of stress-induced depression, linking molecular mechanisms to circuit. In the revised manuscript, we have provided the necessary information in the supplementary data as reviewer suggested to enhance the conclusions.

• *The authors lead into the results by stating that they have data demonstrating that ibotenic acid lesions of CLA leads to depressive like behavior. These data should be shown, at the very least in supplemental figures, or remove the statement. This applies to the other examples of data that is not shown that I highlight below.*

Response: As review suggested, the lesion data has been added as Suppl Fig. 1.

• *Likewise the disconnection experiment is also exciting. The inclusion of those results would nicely increase the rigor of the study.*

Response: The disconnection data has been added in the revised manuscript. The related method has also been provided. Please see Suppl Fig. 7 and page 23, paragraph 3, 4.

• *In lines 129-130, the authors state that CNO was administered 24 hrs after the SDSS but in Fig 1A the diagram shows that CNO was administered prior to SDSS. From the methods, it is unclear when the CNO was administered. These inconsistencies should be clarified.*

Response: We apologized that these necessary experimental conditions were not clearly describe. In SSDS paradigm, the CNO was administrated singly 40 min before SSDS, and 24hr later the animals were subjected to depressive-like behavior testing (Fig. 1a). In CSDS paradigm, the CNO was administrated repeatedly 40 min before each test (Fig. 1h). These conditions have been added in the revised manuscript. Please see Fig. 1 legend.

• *For the chemogenetic experiments in Fig. S2e-i, did the authors administer CNO repeatedly before each test? Or was it only one injection similar to the experiments in Fig 1a-g?*

Response: For the chemogenetic experiments in Fig. S3e-i (originally Fig. S2e-i), CNO was administrated repeatedly 40 min before each test, according to Fig 1 h.

• *The figures were of low resolution in the provided manuscript so some of the data were difficult to judge. For example, for the immunohistochemistry results in Fig. 2, the electrophysiology EPSC data in Fig S4.*

Response: The high-quality figures have been uploaded in the submission system according to the journal requirements.

• *In the electrophysiology data in Fig. S4 the noise in the representative traces is very high and the amplitude of events are very large. Is it possible that the authors recordings biasing towards only detecting large events given the high levels of noise? This may not change the overall interpretation but is a caveat that should be discussed surrounding this finding.*

Response: The high noises may be due to the fact that the EPSCs were highly compressed in the time scale. The typical EPSC events with a three-fold larger than baseline were detected and included in the present work. In the revised manuscript, we have provided a clearer representative trace. Please see Suppl Fig. 5d.

• *The authors report that activity in CLA and PL PV neurons is decreased with fiber photometry experiments but they are comparing activity during different behaviors (social interaction vs tail suspension). Thus, it is difficult to interpret these data. Performing the same behavior in PV neurons would be necessary to be able to draw conclusions across the populations in fiber photometry experiments.*

Response: We thank reviewer for the constructive suggestion. As reviewer suggested, we have provided the data of the activity of GCaMP6s-expressing PL PV interneurons during social interaction. We found that the calcium signals were significantly lower on CSDS mice for PV neurons in the PL. Please see revised Fig. 2 p-r.

• *The authors should also show the cFos data with chemogenetic manipulation of PV neurons in lines 324-330. The description of these data sound like they strengthen the authors conclusions but without seeing the data it is hard to ascertain how much weight is placed on the text.*

Response: As reviewer suggested, we have provided c-fos data with chemogenetic manipulation of PV neurons in the revised manuscript. Please see Suppl Fig. 9.

• *The authors propose that CLA targets deep layers and this may recruit PV neurons in deep layers. Analysis of the of cFos across layers from the authors existing data set may be useful in strengthening this statement and correlating it to cfos activation of pyramidal neurons in a layer-specific manner.*

Response: We agree with reviewer's opinion that c-fos staining data could strengthen our results, however, currently we do not have existing c-fos staining image for further analysis. Jackson with colleagues have examined the topography of CLA axons in the brain, and they found that axons from claustricocortical ChR2-eYFP-labeled neurons were densely innervate all layers of PFC, with a preference for deep layers (Jackson et al., 2018). Xu with colleagues found that PV cells were mainly located in deep layers of PFC (Xu et al., 2010). Thus, we propose that CLA may target PFC deep layer and recruit PV neurons.

• *The authors interpretation that KOR cells preferentially project to PV neurons should be tempered as alternative explanations are possible. The distinct possibilities should be discussed.*

Response: We agree with reviewer's opinion. We have revised our interpretation and discussed distinct possibilities.

"Furthermore, we observed that U50,488H inhibition of CLA-evoked oEPSCs and increased PPR was more profound in PL PV neurons than PNs. A possibility is that CLA neurons that express KOR target both PV interneurons and PNs in the PL, but PV neurons express higher level of KOR. Another possibility is that functional KORs are preferentially targeted to CLA synapses on PV, rather than PNs. Further studies will be necessary to determine the discrete role of dynorphin KOR signaling at projections into these different cell types for controlling the function of CLA-PL pathway and stress-induced depression". Please see page 18, the first paragraph, lines 12-19.

• *The authors should include the nor-BNI data on EPSCs in PV neurons they state is not shown.*

Response: We thank reviewer for the constructive suggestion. The norBNI data has been added. Please see revised Suppl Fig. 10 i-k.

• *For the microinjection procedures the authors should directly state whether microinjections were performed repeatedly in the same mice.*

Response: For microinjection procedure, GABA agonists mixture (baclofen and muscimol) and nor-BNI were given once. GABA antagonists mixture (bicuculline and saclofen) and U50,488H were injected repeatedly. These experimental conditions have been added in the manuscript. Please see method part, page 21, the third paragraph.

• *Quantification of the KOR overexpression images would strengthen establishment of this approach.*

Response: We appreciate reviewer's constructive suggestion, and we have quantified KOR levels

and added in the manuscript as Fig 6g.

• *The results obtained from immunoprecipitation of KOR are interesting but antibodies for KORs, and GPCRs in general, are notoriously non-specific . Appropriate validation of the KOR antibody would be necessary or at the very least discussion of the caveat of not validating this novel antibody would be necessary.*

Response: The KOR antibody PA847 was nicely provided by Dr. Lee-Yuan Liu-Chen lab from Temple University Lewis Katz School of Medicine. The detailed detecting protocol was presented in the method. We previously have validated the efficiency of PA 847 with KOR^{-/-} mice (Zan et al., Cell Rep 2021).

• *The authors can do a better job of covering the literature in the following ways. I suggest the authors carefully vet the way they are citing literature and be sure to incorporate papers that also highlight the discrepancies in the field. Most importantly, the authors ignore the literature that exists in on the PFC Dyn system and cite it's actions elsewhere in the brain.*

• *There are several studies that have studied the dynorphin / KOR system in PFC showing that PFC KOR plays a role in negative affect that the authors need to cite and discuss in the context of the present studies, including Abraham et al 2022, Wall and Messier 2015, Tejada et al 2013, 2015, Bals-Kubik et al 1993, Fassini et al 2015*

• *The authors should cite work from Tejada et al 2013; 2015 and Yarur-Castillo et al 2022 showing that KOR regulates excitatory synapses in the PFC.*

Response: We greatly appreciated reviewer's suggestions, which are very important for improving the quality of our manuscript. All these papers have been cited and discussed in the revised manuscript. Please see page 18, paragraph 1, line 1 to 3; line 11 to 12.

• *The authors also selectively discuss the literature on interneurons in PFC selectively or in inaccurately. Increased*

Response: It is an incomplete statement, but we understand reviewer's concern. We have revised our manuscript, and cited and discussed current discrepancies in the field. Again, we thank reviewer for the constructive suggestions, which are very important for improving the quality of our manuscript.

• *As another example, studies have demonstrated that ketamine and other NMDAR antagonists inhibit PV interneuron activity, counter to how the authors discuss the actions of ketamine in the context of their proposed model. When the authors do mention the NMDAR antagonism mechanisms of ketamine, they do not refer to this vast literature.*

Response: We thank reviewer for pointing this important issue to us. We have revised our discussion by incorporating different papers regarding the discrepancy of ketamine action.

“Our study supports the importance of cortical E/I balance in depression associated with chronic stress. Ketamine was a typical sample for produce fast-acting antidepressant effects, although the mechanisms underlying such effect are not fully elucidated. The counteraction of E/I imbalance may be an important cause, because it can alter E/I imbalance both by directly suppression of excitation of glutamatergic neurons (Zanos and Gould, 2018) and by indirectly potentiation of PV neuronal activity (Chen et al., 2017; Ren et al., 2016; Fushs et al., 2016; Ng et al., 2018), although there is evidence showing that ketamine predominately decreases GABAergic neuronal activity and consequently increases pyramidal cell excitability, downstream of synaptic GABAergic disinhibition (Homayoun and Moghaddam, 2007; Widman and McMahon, 2018). Given that dynorphin/KOR signaling modulates E/I balance in PL micronetwork, our findings may further support that dynorphin/KOR signaling may be a potential target of the development of fast-acting antidepressant agents and the intervention for depression with social stress etiology.

• *Another example is citation of work on SST neurons by Fuchs et al in their discussion of potentiation of PV neurons*

Response: It has already been cited. Please see page 16, paragraph 3, line 11-15.

Minor comments:

• *Line 110-111: the authors state that mice with SSDS displayed vulnerability to subsequent stress in a way that leads the reader to believe this is something they are showing in the present manuscript. Please update the language accordingly to say this is something established in the literature.*

Response: It has been revised as reviewer suggested. Please see page 5, the first paragraph, line 9.

Reference

Chen, C.C., Lu, J., Yang, R., Ding, J.B., and Zuo, Y. (2017). Selective activation of parvalbumin interneurons prevents stress-induced synapse loss and perceptual defects. *Molecular Psychiatry* 23:7 23, 1614–1625.

Fuchs, T., Jefferson, S.J., Hooper, A., Yee, P.H., Maguire, J., and Luscher, B. (2016). Disinhibition of somatostatin-positive GABAergic interneurons results in an anxiolytic and antidepressant-like brain state. *Molecular Psychiatry* 22:6 22, 920–930.

Homayoun, H. & Moghaddam, B. (2007) NMDA receptor hypofunction produces opposite effects on prefrontal cortex interneurons and pyramidal neurons. *J Neurosci* 27, 11496–11500.

Jackson J., Karnani, M.M., Zemelman, B.V., Burdakov, D., Lee, A.K. (2018) Inhibitory Control of Prefrontal Cortex by the Claustrum. *Neuron* 99, 1029–1039

Ng, L.H.L., Huang, Y., Han, L., Chang, R.C.C., Chan, Y.S., and Lai, C.S.W. (2018). Ketamine and

selective activation of parvalbumin interneurons inhibit stress-induced dendritic spine elimination. *Translational Psychiatry* 8:18, 1–15.

Ren, Z., Pribrag, H., Jefferson, S.J., Shorey, M., Fuchs, T., Stellwagen, D., and Luscher, B. (2016). Bidirectional Homeostatic Regulation of a Depression-Related Brain State by Gamma-Aminobutyric Acidergic Deficits and Ketamine Treatment. *Biol Psychiatry* 80, 457–468.

Widman, A. J. & McMahon, L. L. (2018) Disinhibition of CA1 pyramidal cells by lowdose ketamine and other antagonists with rapid antidepressant efficacy. *Proc. Natl Acad. Sci. USA* 115, 3007–3016.

Zan, G. Y., Wang, Y. J., Li, X. P., Fang, J. F., Yao, S. Y., Du, J. Y., Wang, Q., Sun, X., Liu, R., Shao, X. M., Long, J. D., Chai, J. R., Deng, Y. Z., Chen, Y. Q., Li, Q. L., Fang, J. Q., Liu, Z. Q. & Liu, J. G. (2021). Amygdalar κ -opioid receptor-dependent upregulating glutamate transporter 1 mediates depressive-like behaviors of opioid abstinence. *Cell Rep* 37, 109913.

Zanos, P., Gould, T.D. (2018) Mechanisms of Ketamine Action as an Antidepressant. *Mol Psychiatry* 23(4), 801–811.

Reviewer #3 (Remarks to the Author):

In this study, Wang and colleagues investigate the molecular and circuit mechanisms of stress-induced disruption of cortical micro-network. The authors showed that the CSDS decreased the output of CLA-PL glutamatergic circuit. Optogenetic manipulation of this circuit bidirectionally regulated the depression-related behaviors induced by CSDS. Their further experiments provide the molecular mechanism that dynorphin/KOR signaling in CLA-PL circuit was involved in the depression-related behaviors. Overall, this is a very interesting and well-completed study. The experiments used elegant viral approaches, are well designed and executed to explore the circuit synaptic connection and function. There are some minor comments that should be addressed before publication.

The findings on dynorphin are an important part for this study. Some dynorphin results should be moved from supplementary figure 8 to main figure 6.

Response: As reviewer suggested, the dynorphin data has been moved to Fig 6. Please see revised Fig 6. a-d.

In Figure 3, it is confusing that different colors are used for groups. It would be better to use one color for all ChR2 groups, and another for all ArchT groups.

Response: The Figures have been revised. Please see revised Fig. 3.

The labels could be better in Figure 5. It looks like CSDS did not work. Actually, all black bars in the bar graph should be ChR2 groups or ArchT groups, and other colored bars should be ChR2 + CNO groups or ArchT + CNO groups.

Response: The Figures have been revised. Please see revised Fig. 5.

In the figure legend for Figure 2, the label (-180 s - 180 s) may change to (-180 s to +180 S); similarly, (-60 s - 380 s) to (-60 s to + 380). In addition, it is unclear why the recording ranges are

different between the social interaction (panel 2d and 2j) and tail suspension test (panel 2p). Related information should be provided.

Response: we have provided the data of the activity of GCaMP6s-expressing PL PV interneurons during social interaction, as reviewer suggested. Please see revised Fig 2. p-r.

In the abstract, “being critical for 35 depression in mice” should be rephrased. It would be better to use depression-related behaviors in mice.

Also in the abstract, “and possibly intervening depression by targeting CLA-PL circuit” needs to be rewritten. The sentence is not complete.

Response: They have been revised as reviewer suggested. Please see abstract section.

Move CLA label to the location of LCA in Figure 1b and 1i.

Response: It has been revised as reviewer suggested.

More information about the neurogranin should be included, to explain why the authors used neurogranin antibody to label those neurons.

Response: Neurogranin (NG) was used as pyramidal neurons marker (Wang et al., 2021). We have added related description in the revised manuscript. Please see Fig. 2 legend.

In the figure legend of Figure 3, “regulate mice depression-like behaviors” may change to “regulate depression-like behaviors of mice”.

Response: It has been changed as reviewer suggested. Please see revised Fig 3 legend.

Reference:

Wang, M., Gallo N.B., Tai Y., Li B., Van Aelst L. (2020). Oligophrenin-1 moderates behavioral responses to stress by regulating parvalbumin interneuron activity in the medial prefrontal cortex. *Neuron* 109(10): 1636–1656.

REVIEWERS' COMMENTS

Reviewer #1 (Remarks to the Author):

The authors' responses are generally thorough, and we are satisfied with the changes made in the manuscript and figures. We have some minors for reference only.

Reviewer's original comments: The Co-localization of GAD65 and mCherry in Supplementary Figure 2 is concerning. It is impossible to ascertain that the CNO-stimulated population in the experiment is glutamatergic. The CLA contains 85% glutamatergic and 10-15% GABA-ergic neurons, so the chosen promoter is not selective for glutamatergic neurons and will infect the cells unbiasedly.

Author Response: In this study, the chosen promoter is CaMKII α , which is a known marker of excitatory neurons. Previous studies have shown that roughly 85% of claustral projection neurons are glutamatergic with expressing gene encoding vGlut2 (Mathur, 2014; Gomez-Urquijo et al., 2000). Ideally AAV expressing mCherry-tagged Gi or Gq DREADD under the control of CaMKII α should be restricted to glutamatergic neurons, however, we found that 85% colocalization of mCherry and vGlut1, 9% colocalization of mCherry and GAD65. It could be due to leaky virus expression or off-target infection. Several studies have found that under many conditions in both responder mouse lines and with helper viruses, there is the potential for low levels of "leak" expression (Seidler et al., 2008; Wall et al., 2010; Miyamichi et al., 2013). To better address reviewer's concern, we conducted additional experiments, and we still found that virus containing a CaMKII α promoter predominantly expressed in glutamatergic neurons in the CLA, with most (90.36%) mCherry+ neurons coexpressed with vGlut1+ mRNA, a small fraction (9.64%) expressing vGlut1-.

Reviewer: We appreciate the authors' use of additional experiments to address this concern. However, we are unsure if removing previously included data from Supplementary Figure 2 regarding the GAD65 expression is a good idea. We recommend that the data be re-introduced to the Supplementary Figures, and that the potential for GABAergic interneuron contributions be discussed in the manuscript. While the results may be attributed to "leaky expression," as the authors explained, the possibility of GABAergic interneuron contribution still cannot be ruled out with the data provided. It remains a possibility that some of the neurons targeted in these experiments were not glutamatergic, and GABAergic interneurons may be partially responsible for driving the behavioral changes described.

Reviewer #2 (Remarks to the Author):

The authors have addressed my concerns

Reviewer #3 (Remarks to the Author):

The authors have addressed my original concerns and comments thoroughly by including additional information or moving the data from supplementary to main figure. No further comments.

Responses to reviewer's comments

We greatly appreciate reviewers' efforts in reviewing our manuscript.

REVIEWERS' COMMENTS

Reviewer #1 (Remarks to the Author):

The authors' responses are generally thorough, and we are satisfied with the changes made in the manuscript and figures. We have some minors for reference only.

Reviewer's original comments: The Co-localization of GAD65 and mCherry in Supplementary Figure 2 is concerning. It is impossible to ascertain that the CNO-stimulated population in the experiment is glutamatergic. The CLA contains 85% glutamatergic and 10-15% GABA-ergic neurons, so the chosen promoter is not selective for glutamatergic neurons and will infect the cells unbiasedly.

Author Response: In this study, the chosen promoter is CaMKII α , which is a known marker of excitatory neurons. Previous studies have shown that roughly 85% of claustral projection neurons are glutamatergic with expressing gene encoding vGlut2 (Mathur, 2014; Gomez-Urquijo et al., 2000). Ideally AAV expressing mCherry-tagged Gi or Gq DREADD under the control of CaMKII α should be restricted to glutamatergic neurons, however, we found that 85% colocalization of mCherry and vGlut 1, 9% colocalization of mCherry and GAD65. It could be due to leaky virus expression or off-target infection. Several studies have found that under many conditions in both responder mouse lines and with helper viruses, there is the potential for low levels of "leak" expression (Seidler et al., 2008; Wall et al., 2010; Miyamichi et al., 2013). To better address reviewer's concern, we conducted additional experiments, and we still found that virus containing a CaMKII α promoter predominantly expressed in glutamatergic neurons in the CLA, with most (90.36%) mCherry+ neurons coexpressed with vGlut1+ mRNA, a small fraction (9.64%) expressing vGlut1—.

Reviewer: We appreciate the authors' use of additional experiments to address this concern. However, we are unsure if removing previously included data from Supplementary Figure 2 regarding the GAD65 expression is a good idea. We recommend that the data be re-introduced to the Supplementary Figures, and that the potential for GABAergic interneuron contributions be discussed in the manuscript. While the results may be attributed to "leaky expression," as the authors explained, the possibility of GABAergic interneuron contribution still cannot be ruled out with the data provided. It remains a possibility that some of the neurons targeted in these experiments were not glutamatergic, and GABAergic interneurons may be partially responsible for driving the behavioral changes described.

Response: We appreciate reviewer's comments, which are very important for improving the quality of our manuscript. As reviewer suggested, the original Suppl Fig 2 has been re-introduced into the manuscript. Please see revised Suppl Fig 3, page 5, paragraph 2, line 3-5.

However, we respectively disagree that the claustrum GABAergic interneurons may be involved in this work because it lacks of supporting data. In contrast, all the results we have including c-fos staining, in situ hybridization, whole-cell electrophysiological recording, optogenetics, chemogenetics and fiber photometry suggest the importance of claustrum excitatory neurons that project to prelimbic cortex and play a key role in driving depressive-like behaviors after social defeat stress.